# Variational Learning of Disentangled Representations

**Yuli Slavutsky** [* 1]   **Ozgur Beker** [* 2]   **David M. Blei** [1 3]   **Bianca Dumitrascu** [2 1 3 4]

## Abstract

Disentangled representations separate factors that are shared across conditions from those that are condition-specific. Such separation is needed for generalization to new domains, treatments, patients, or species. A dominant line of work pursues this goal through variational formulations. While these approaches achieve partial disentanglement, they often exhibit three common limitations: they either do not remove all condition-specific information from the condition-specific representation, allow the condition-specific representation to become uninformative, or impose independence assumptions that do not reflect the underlying generative process. In this work, we introduce DisCoVR, a variational framework that addresses these limitations. Its objective is aligned with the probabilistic structure of the data-generating process, and includes an adversarial term that prevents condition-specific information from being encoded in the condition-specific representation. DisCoVR reconstructs the data from both shared and condition-specific representations, ensuring that each remains informative, and uses a structured prior that further reinforces the informativeness of both representations. We show that across synthetic, image, and single-cell RNA-sequencing datasets, DisCoVR achieves stronger disentanglement compared to previous approaches.

[1]Department of Statistics, Columbia University, New York City, USA [2]Irving Institute for Cancer Dynamics, Columbia University, New York City, USA [3]Department of Computer Science, Columbia University, New York City, USA [4]Columbia Stem Cell Initiative, Columbia University, New York City, USA. Correspondence to: Bianca Dumitrascu <bmd2151@columbia.edu>, Yuli Slavutsky <ys3938@columbia.edu>.

*Proceedings of the $43^{rd}$ International Conference on Machine Learning*, Seoul, South Korea. PMLR 306, 2026. Copyright 2026 by the author(s).

## 1. Introduction

Neural network–based models excel at learning rich representations of complex data, and are increasingly used in settings where each observation $x \in \mathcal{X} \subseteq \mathbb{R}^d$ is paired with a condition label $y \in \{1, \dots, K\}$, such as patient, site, or experimental condition. Generalizing these representations to new domains often requires disentangling factors shared across conditions from those specific to each $y$.

Generative models provide a natural framework for uncovering latent structure and learning data representations, with prominent examples including Generative Adversarial Networks (GANs) (Goodfellow et al., 2020), Variational Autoencoders (VAEs) (Kingma & Welling, 2014), and diffusion models (Sohl-Dickstein et al., 2015; Ho et al., 2020). Among these, VAEs and their extensions are particularly well-suited to transfer learning and domain adaptation (Akrami et al., 2020; Lovrić et al., 2021; Godinez et al., 2022; Zhang et al., 2023), thanks to their probabilistic foundation and ability to capture uncertainty.

Accordingly, several VAE-based methods have been proposed to integrate data across multiple conditions or sources (Xu et al., 2021; Lotfollahi et al., 2019; Boyeau et al., 2022), but only a few explicitly disentangle invariant and condition-specific components (Sohn et al., 2015; Klys et al., 2018; Joy et al., 2020; Ilse et al., 2020).

While these approaches improve separation to some degree, they either (i) leave the shared component free to retain label information without an explicit mechanism enforcing invariance (Sohn et al., 2015; Ilse et al., 2020), (ii) reconstruct $x$ jointly from the invariant and condition-specific components, so that the invariant one can remain uninformative (Klys et al., 2018), (iii) impose independence assumptions that misalign with the underlying generative structure (Ilse et al., 2020), or (iv) encode in their priors the assumption that the invariant representation in fact can vary with the label (Joy et al., 2020).

In this work, we introduce a framework for learning *disentangled representations in multi-condition datasets* that addresses these limitations: (a) We formulate a principled probabilistic objective that encodes the correct conditional independencies. (b) We specify a prior structure in which the condition-specific representation $w$ depends on the mean

of the invariant representation $z$. Since $w$ depends only on class-specific aggregation of $z$, the invariant representation cannot absorb condition-specific leakage from $w$, while forcing $z$ to remain informative. (c) Our method uses reconstruction paths from both representations, which further enforces informativeness of $z$. (d) The resulting optimization objective includes an adversarial term that explicitly discourages leakage of condition-specific information into the invariant representation.

Our approach formulates disentanglement as a max–min game, which we show to admit a unique equilibrium. Empirically, we show through extensive experiments on synthetic benchmarks and real-world datasets, that our method consistently improves over existing approaches in disentangling shared and condition-specific structure.

## 2. DisCoVR: Disentangling Common and Variant Representations

For the task of learning disentangled representations from multi-condition data, we consider a dataset $\mathcal{D} = \{(x_i, y_i)\}_{i=1}^n$ consisting of inputs $x_i \in \mathcal{X} \subseteq \mathbb{R}^d$ collected from associated condition labels $y_i \in \{1, \ldots, K\}$. For each class $k$ (corresponding to a study or experimental condition), the associated subset $\mathcal{D}_k := \{x_i : y_i = k\}$ consists of i.i.d. samples drawn from a class-conditional distribution $p(x \mid y = k)$.

### 2.1. Model assumptions

We assume that the data is generated by latent variables $z$ and $w$, such that the joint distribution $p(x, y, z, w)$ factorizes according to the probabilistic graphical model illustrated in Figure 1, i.e.,

$$p(x, y, z, w) = p(y)\, p(w \mid y)\, p(z)\, p(x \mid z, w). \quad (1)$$

This model encodes two key conditional independence assumptions:

1. *Latent variable conditional independence:* Given the condition $y$, the latent representations $z$ and $w$ are conditionally independent: $z \perp w \mid y$.

2. *Sufficiency of the condition-aware latent representation:* The input $x$ is conditionally independent of the condition $y$ given $w$: $x \perp y \mid w$.

Note that in this formulation, $z$ and $w$ are no longer independent if conditioned also on $x$, that is, $z \not\perp w \mid x, y$.

### 2.2. Target posterior structure

In our model, each observation $x$ is generated from two latent variables: $z$, which is *condition-invariant*, and $w$, which

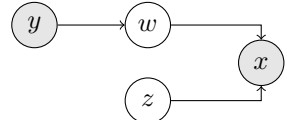

*Figure 1.* Probabilistic graphical model: gray circles denote observed variables, white circles show latent variables.

is *condition-aware* through its dependence on $y$. Our goal is to learn probabilistic representations where the marginals of $z$ and $w$ preserve this structure, yielding disentangled factors. That is, we aim to approximate the posterior $p_{z,w|x,y}$.

However, approximating the full posterior with a variational distribution $q_{z,w|x,y}$ is intractable: even for simple variational families such as Gaussians, modeling the dependencies between $z$ and $w$ requires a full covariance structure, which is computationally prohibitive in high dimensions. To mitigate this, we employ a factorized approximation $q_{z|x}\, q_{w|x,y}$.

Our variational approximation is guided by two complementary objectives: (i) $q_{z|x}$ closely approximating the marginal posterior $p_{z|x}$; and (ii) the product $q_{z|x}\, q_{w|x,y}$ closely approximating the true posterior $p_{z,w|x,y}$. Formally, given variational families[1] $\mathcal{Q}_z$ and $\mathcal{Q}_w$ we seek to find $q_{z|x}^* \in \mathcal{Q}_z$ and $q_{w|x,y}^* \in \mathcal{Q}_w$ that minimize the following sum of Kullback-Leibler (KL) divergences:

$$q_{z|x}^*,\, q_{w|x,y}^* = \arg \min_{\substack{q_{z|x} \in \mathcal{Q}_z \\ q_{w|x,y} \in \mathcal{Q}_w}} \big[ \mathcal{D}_{\mathrm{KL}}(q_{z|x} \,\|\, p_{z|x}) \quad (2)$$
$$+ \mathcal{D}_{\mathrm{KL}}(q_{z|x}\, q_{w|x,y} \,\|\, p_{z,w|x,y}) \big].$$

### 2.3. Optimization objective

Since direct evaluation of the KL divergences in Equation 2 is intractable, we optimize a surrogate objective consisting of two ELBO terms.

The corresponding ELBO objective to minimize $\mathcal{D}_{\mathrm{KL}}(q_{z|x} \,\|\, p_{z|x})$ is

$$\mathcal{L}_z(q_{z|x}, p; x) := \mathbb{E}_{q_{z|x}} \left[ \log p(x \mid z) \right] - \mathcal{D}_{\mathrm{KL}}(q_{z|x} \,\|\, p_z), \quad (3)$$

and the ELBO objective for the second KL term, $\mathcal{D}_{\mathrm{KL}}(q_{z|x} q_{w|x,y} \,\|\, p_{z,w|x,y})$, is

$$\mathcal{L}_w(q_{w|x,y}, p; x, y) := \mathbb{E}_{q_{z|x}} \big[ \mathbb{E}_{q_{w|x,y}} [\log (p(x|z, w))] \big]$$
$$- \mathcal{D}_{\mathrm{KL}} \left( q_{z|x} \,\|\, p_z \right) - \mathcal{D}_{\mathrm{KL}} \left( q_{w|x,y} \,\|\, p_{w|y} \right). \quad (4)$$

Note that $\mathcal{L}_w(q_{w|x,y}, p; x, y)$ is the ELBO objective that corresponds to a factorized posterior $q_{z|x} q_{w|x,y}$. In Proposition 2.1 we examine the gap between this objective and an ELBO

---

[1]Here we consider general families and specify our concrete choices in §2.4.

corresponding to a full variational posterior. This can be interpreted as the cost of enforcing a condition-invariant latent representation, specifically, constraining $z$ to depend only on $x$.

**Proposition 2.1.** *For random variables $x, y, z, w$ following the graphical model in Figure 1,*

$$\text{ELBO}(q, p; x, y) - \mathcal{L}_w(q_{w|x,y}, p; x, y)$$
$$= \mathbb{E}_{q_{w|x,y}} \left[ KL \left( q_{z|x} \,\|\, p_{z|w,x,y} \right) \right]$$

*where*

$$\text{ELBO}(q, p; x, y) := \log p(x \mid y) - \mathcal{D}_{\text{KL}} \left( q_{w|x,y} \,\|\, p_{w|x,y} \right).$$

The proof is provided in Appendix B.1.

Note that a full definition of the objectives in Equations 3 and 4 requires the specification of corresponding prior distributions, namely $p_z$ and $p_{w|y}$. We defer their definitions to §2.4.

Equation 4 provides an evidence lower bound on the conditional log-likelihood $\log p(x \mid y)$. By adding $\log p(y)$, this bound extends to the joint log-marginal likelihood $\log p(x, y)$. Beyond optimizing this objective, we aim to ensure that the marginal distribution over $y$ implicitly induced by the latent representations is consistent with the true $p(y)$.

To this end, we introduce an auxiliary classifier $g(y \mid z)$ as a form of posterior regularization (Ganchev et al., 2010). This classifier captures the residual predictive signal about $y$ in $z$ and is trained by minimizing the expected negative log-likelihood $-\mathbb{E}_{q(z|x)} \log g(y \mid z)$. If $z$ is truly independent of $y$, then $g(y \mid z)$ will approximate the marginal distribution $p(y)$. By penalizing deviations from this behavior, we enforce the structural constraint $z \perp y$ in the learned representation.

For this term to effectively encourage $q_{z|x}$ to discard condition-specific information, the classifier $g_{y|z} \in \mathcal{G}$ must be trained adversarially, with its own update step. This prevents degenerate solutions in which the loss is minimized without removing information about $y$ from $z$, for example, by collapsing $g$ to a constant predictor that ignores its input.

Combining the three terms above, we define the objective

$$\mathcal{L}(q_{z|x}, q_{w|x,y}, g_{y|z}; x, y) \tag{5}$$
$$= \mathcal{L}_z(q_{z|x}, p; x) + \mathcal{L}_w(q_{w|x,y}, p; x, y) - \mathbb{E}_{q_{z|x}} \log g(y \mid z),$$

which can be explicitly expressed as

$$\mathcal{L}(q_{z|x}, q_{w|x,y}, g_{y|z}; x, y) := \mathbb{E}_{q_{z|x}} [\log p(x \mid z)] \tag{6}$$
$$+ \mathbb{E}_{q_{z|x}} \left[ \mathbb{E}_{q_{w|x,y}} [\log p(x \mid z, w)] \right] - \mathbb{E}_{q_{z|x}} [\log g(y \mid z)]$$
$$- 2\mathcal{D}_{\text{KL}}(q_{z|x} \,\|\, p_z) - \mathcal{D}_{\text{KL}} \left( q_{w|x,y} \,\|\, p_{w|y} \right).$$

Finally, to enable flexible trade-offs between reconstruction expressiveness and disentanglement, we introduce weighting terms $\alpha = (\alpha_1, \alpha_2)$ into the objective following the motivation of $\beta$-VAEs (Higgins et al., 2017):

$$\mathcal{L}_\alpha(q_{z|x}, q_{w|x,y}, g_{y|z}; x, y) := \mathbb{E}_{q_{z|x}} [\log p(x \mid z)] \tag{7}$$
$$+ \mathbb{E}_{q_{z|x}} \left[ \mathbb{E}_{q_{w|x,y}} [\log p(x \mid z, w)] \right] - \mathbb{E}_{q_{z|x}} \log g(y \mid z)$$
$$- \alpha_1 \mathcal{D}_{\text{KL}}(q_{z|x} \,\|\, p_z) - \alpha_2 \mathcal{D}_{\text{KL}} \left( q_{w|x,y} \,\|\, p_{w|y} \right).$$

Accordingly, the mean weighted objective is suitable for max-min optimization of the form:

$$\tag{8}$$

$$\max_{q_{z|x} \in \mathcal{Q}_z} \max_{q_{w|x,y} \in \mathcal{Q}_w} \min_{g_{y|z} \in \mathcal{G}} \mathbb{E}_{p_{x,y}} \left[ \mathcal{L}_\alpha(q_{z|x}, q_{w|x,y}, g_{y|z}; x, y) \right].$$

## 2.4. Latent prior models and variational approximations

**Prior specification** We place a standard Normal prior over the condition-invariant latent variable, $p_z = \mathcal{N}(0, I)$, which reflects a non-informative prior belief over its values.

For the condition-aware latent variable $w$, we define a class-conditional Gaussian prior $p_{w|y}$. As a simple choice, we let $w$ have the same dimensionality as $z$ and specify

$$p(w \mid y = k) = \mathcal{N}(\mu_k, I), \quad \mu_k := \mathbb{E}_{p_{x|y=k}} \left[ \mathbb{E}_{q_{z|x}} [z] \right]. \tag{9}$$

Here $\mu_k$ is the mean of the inferred latent representations $z$ within the $k$-th class[2].

This specification induces a coupling between the two latent variables through the data distribution. Aligning $p_{w|y}$ with the class-wise expectations of the invariant variable, further encourages $q_{z|x}$ to encode informative representations, since capturing the shared structure will now not only increase $\mathcal{L}_z(q_{z|x}, p; x)$, but also decrease $\mathcal{D}_{\text{KL}} \left( q_{w|x,y} \,\|\, p_{w|y} \right)$, and as a result increase $\mathcal{L}_w(q_{w|x,y}, p; x, y)$.

However, this coupling between $z$ and $w$ is not defined at the level of individual samples: The prior for $w$ depends on $z$ only through the class-wise mean $\mu_y = \mathbb{E}_{p(x|y)} \mathbb{E}_{q(z|x)}[z]$, not through a given $z$. Similarly, $q_{z|x}$ never conditions on $w$ or $y$, so the invariant representation cannot absorb condition-specific leakage from $w$.

Importantly, for a truly condition-agnostic $q_{z|x}$, the conditional expectations $\mu_k$ will collapse to a shared mean $\mu := \mathbb{E}_{p_x} \left[ \mathbb{E}_{q_{z|x}} [z] \right]$. In this case $p_{w|y}$ becomes a shared prior across classes, centered at a meaningful point in the latent space, rather than an uninformative one.

As the following proposition establishes, this anchoring of the prior $p_{w|y}$ in the variational distribution $q_{z|x}$ preserves

---

[2]Similarly, if $z$ and $w$ have different dimensions, the mean aggregation can be replaced with a neural network that maps the inferred representations $z$ for each class to the parameters of the Gaussian prior.

the convex–concave structure of the objective, ensuring that the resulting max-min problem has a unique optimal solution.

**Proposition 2.2.** *Let $\mathcal{Q}_z$ and $\mathcal{Q}_w$ be convex parametric families of variational distributions over $z$ and $w$, respectively, and let $\mathcal{G}$ denote a convex set of classifiers such that $g(x) \in [0,1]$ for all $g \in \mathcal{G}$. Assume the latent priors are given by $z \sim p(z)$ and $p(w|y) = \mathcal{N}(\mu_y, I)$, where $p(z)$ is a continuous strictly positive distribution, and $\mu_y = \mathbb{E}_{p_{x|y}} \left[ \mathbb{E}_{q_{z|x}}[z] \right]$. Then, under standard regularity conditions (see Appendix B.2.1), there exists a unique saddle point:*

$$\left( q^*_{z|x}, q^*_{w|x,y}, g^*_{y|z} \right)$$
$$= \max_{q_{z|x} \in \mathcal{Q}_z} \max_{q_{w|x,y} \in \mathcal{Q}_w} \min_{g_{y|z} \in \mathcal{G}} \mathcal{L}(q_{z|x}, q_{w|x,y}, g_{y|z}).$$

The proof is provided in Appendix B.2.2.

**Specification of variational families**  We set both variational families $\mathcal{Q}_z$ and $\mathcal{Q}_w$ as $d$-dimensional Gaussian distributions with diagonal covariance matrices. Accordingly, each variational distribution is parameterized by a mean vector $\mu \in \mathbb{R}^d$ and a vector of variances $\sigma^2 \in \mathbb{R}^d_+$ corresponding to the diagonal of the covariance matrix, yielding $\theta = (\mu, \sigma^2)$.

## 3. Encoder-decoder model

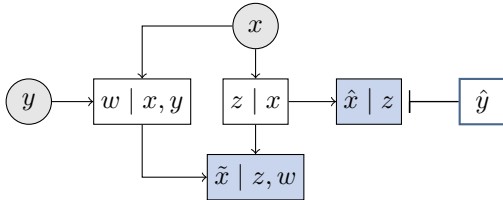

*Figure 2.* Encoder–decoder architecture: the inhibition arrow from $\hat{y}$ to $\hat{x}$ corresponds to the adversarial component.

In order to optimize the objective in Equation 8 with respect to $q_{z|x}$, $q_{w|x,y}$, and $g_{y|z}$ over the dataset $\mathcal{D}$, we introduce an encoder-decoder framework (illustrated in Figure 2). In this framework, two separate reconstructions of $x$ are generated: one, denoted $\hat{x} \sim p_{x|z}$, where $z$ is sampled from the condition-invariant posterior $q_{z|x}$, and the other, denoted $\tilde{x} \sim p_{x|z,w}$, where in addition, $w$ is sampled from the condition-aware posterior $q_{w|x,y}$. The corresponding procedure is summarized in Algorithm 1.

**Condition-agnostic representation**  An input $x \in \mathcal{X}$ is mapped to the variational parameters $\theta_z = (\mu_z, \sigma^2_z)$ by an encoder neural network $f^z_\phi : \mathcal{X} \to \mathbb{R}^d \times \mathbb{R}^d_+$ parametrized by weights $\phi$. A latent encoding $z \sim q_{\theta_z}$ is then sampled and mapped to a reconstruction $\hat{x}$ via a decoder neural network $h^z_\psi : \mathbb{R}^d \to \mathcal{X}$ parametrized by weights $\psi$.

**Adversarial classifier**  Instead of training a high-capacity classifier directly from $z$ to $y$, we use the reconstruction $\hat{x}$ from $z$, and predict $y$ from $\hat{x}$ via a simpler model $g_\beta : \hat{\mathcal{X}} \to [0,1]^K$ (in most cases, implemented as a multinomial logistic regression with class-specific weights $\beta = \beta_k{}^K_{k=1}$, or a shallow MLP). Since $\hat{x}$ is a deterministic function of $z$, this is equivalent to applying a restricted classifier on $z$. By the data processing inequality, such a classifier can only capture a subset of the information $z$ contains about $y$; as a result, maximizing this lower bound on $I(z; y)$ also maximizes $I(z; y)$ itself. Although this substitution weakens the estimation of the cross-entropy term $-\mathbb{E}_{q_{z|x}} \log g(y \mid z)$ from an information-theoretic standpoint, we observed this to be often advantageous in practice: predicting $y$ from $\hat{x}$ reduces the variance introduced by sampling $z \sim q_{\theta_z}$, providing a regularizing effect that prevents $q_{\theta_z}$ from overfitting to noisy classifier signals. We empirically evaluate this design choice in Appendix F, where we compare it to a classifier operating directly on $z$.

**Condition aware representation**  A labeled input pair $(x, y) \in \mathcal{X} \times \{1, \ldots, K\}$ is mapped to the parameters $\theta_w = (\mu_w, \sigma^2_w)$ using an encoder neural-network $f^w_\rho : \mathcal{X} \times \{1, \ldots, K\} \to \mathbb{R}^d \times \mathbb{R}^d_+$ parametrized by weights $\rho$. A sample $w \sim q_{\theta_w}$ is then drawn, and the pair $(z, w)$ is mapped to a reconstruction $\tilde{x}$ via a decoder neural-network $h^{z,w}_\eta : \mathbb{R}^{d_z + d_w} \to \mathcal{X}$, parametrized by weights $\eta$. To compute the prior $p_{w|y}$, we estimate the class-specific mean as $\hat{\mu}_k := \frac{1}{n_k} \sum_{i: y_i = k} z_i$ where each $z_i \sim q(z \mid x_i)$ is sampled from the encoder given an input $x_i$ with label $y_i = k$, and $n_k$ is the number of training points with the label $y = k$.

In practice, the idealized game in Proposition 2.2 is implemented with the standard relaxations used in VAE-based models. Specifically, we use a single-sample Monte Carlo estimate to approximate the expectations in Equation 8. Instead of directly sampling from $q_\theta$, we employ the reparameterization trick to enable differentiable sampling: we sample $\epsilon \sim \mathcal{N}(0, I)$ and obtain a sample from $q_\theta$ by applying a deterministic transformation of $\epsilon$ based on the variational parameters $\theta$.

## 4. Comparison to previous approaches

Here we review VAE-based methods for disentangled representation learning, which form the primary basis for comparison with our approach. Broader related literature is discussed in Appendix A.

VAEs (Kingma & Welling, 2014) are generative models that learn latent representations by maximizing the evidence lower bound (ELBO) on the data log-likelihood:

$$\mathbb{E}_{q_{z|x}}[\log p(x \mid z)] - \mathcal{D}_{\mathrm{KL}} \left( q_{z|x} \parallel p_z \right) \leq \log p(x),$$

where $(x, z) \sim p$, and $z|x \sim q$ is a latent variable inferred

**Algorithm 1**

**Input:** Data $\mathcal{D} = \{x_{1:n}, y_{1:n}\}$, number of training iterations $J$, initial parameters $\phi^{(0)}, \psi^{(0)}, \rho^{(0)}, \eta^{(0)}, \beta^{(0)}$, learning rates $\gamma_1, \gamma_2$, weighting terms $\alpha = (\alpha_1, \alpha_2)$

**for** $1 \leq j \leq J$ **do**

Compute $\theta_z = f^z_{\phi^z}(x)$ and $\theta_w = f^w_{\rho^{(j-1)}}(x, y)$

Sample condition invariant and aware latent variables $z \sim q_{\theta_z}$ and $w \sim q_{\theta_w}$

Compute reconstructions $\hat{x} = h^z_{\psi^{(j-1)}}(z)$ and $\tilde{x} = h^{z,w}_{\eta^{(j-1)}}(z, w)$

Compute condition prediction $\hat{y} = g_{\beta^{(j-1)}}(\hat{x})$

Update classifier parameters:

$$\beta^{(j)} \leftarrow \beta^{(j-1)} - \gamma_1 \nabla_\beta \, \mathcal{L}_\alpha(q_{z|x}, q_{w|x,y}, g_{y|z})$$

with the gradient evaluated at

$$\Omega^{(j-1)} := \left( \phi^{(j-1)}, \psi^{(j-1)}, \rho^{(j-1)}, \eta^{(j-1)} \right).$$

Update encoder and decoder parameters $\Omega^{(j)}$

$$\Omega^{(j)} \leftarrow \Omega^{(j-1)} + \gamma_2 \nabla_{\phi,\psi,\rho,\eta} \, \mathcal{L}_\alpha(q_{z|x}, q_{w|x,y}, g_{y|z})$$

with the gradient evaluated at $\beta^{(j)}$.

**end for**

**Return:** $\beta^{(J)}, \Omega^{(J)} = \left( \phi^{(J)}, \psi^{(J)}, \rho^{(J)}, \eta^{(J)} \right).$

---

from data. VAEs consist of an encoder $q_{z|x}$ that maps inputs to latent distributions, and a decoder $p_{x|z}$ that reconstructs inputs from latent representations. The learning process frames posterior inference as KL-regularized optimization over a variational family $\mathcal{Q}$, aiming to approximate the posterior $p_{z|x}$ under a typically simple prior $p(z)$. Several VAE extensions were proposed to encourage disentanglement. These are discussed below.

**Conditional VAEs** (Sohn et al., 2015) incorporate supervision into the standard VAE model by conditioning both the encoder and decoder on an observed label $y$, yielding the following objective:

$$\mathbb{E}_{q_{z|x,y}}[\log p(x|z,y)] - \mathcal{D}_{\text{KL}}\left(q(z|x,y)\|p(z)\right).$$

While this allows controlled generation and partial disentanglement between $z$ and $y$, since the prior $p(z)$ is global (e.g. $\mathcal{N}(0, I)$) and $z$ is inferred from both $x$ and $y$, no mechanism forces $z$ to discard label information. On the contrary, encoding both $x$-specific and $y$-specific information in $z$ will improve reconstruction error.

**Conditional Subspace VAEs** (CSVAEs) (Klys et al., 2018), explicitly factorize the latent space into a shared component $z$ and a label-specific component $w$ (see Supplementary Figure 1a). Similarly, their hierarchical extension (Beker et al.,

2024) introduces an intermediate latent variable between $x$ and $(z, w)$. As in our approach, to encourage disentanglement, CSVAEs introduce an adversarial regularization term that penalizes mutual information between $z$ and $y$, thereby discouraging predictability of $y$ from $z$. They are learned by optimizing

$$\mathbb{E}_{q_{z,w|x,y}}[\log p(x \mid w, y)] - \mathbb{E}_{q_{z|x}}\left[ \int g(y \mid z) \log g(y \mid z) \, dy \right]$$
$$- \mathcal{D}_{\text{KL}}\left( q_{w|x,y} \,\|\, q_{w|y} \right) - \mathcal{D}_{\text{KL}}\left( q_{z|x} \,\|\, p_z \right).$$

However, here the reconstruction $p(x \mid w, y)$ uses only the condition-specific representation $w$, and therefore may result in uninformative invariant representation $z$.

**Domain Invariant VAEs (DIVA)** (Ilse et al., 2020), shown in Supplementary Figure 1b, introduce two latent variables, $z$ and $w$, where $w$ captures label-related features by jointly optimizing a classifier $q(y \mid w)$ alongside the remaining objective. For fully supervised cases, the DIVA model optimizes

$$\mathbb{E}_{q_{z,w|x}}[\log p(x \mid z, w)] + \mathbb{E}_{q_{w|x}}[\log q(y \mid w)]$$
$$- \mathcal{D}_{\text{KL}}\left( q_{z|x}\|p_z \right) - \mathcal{D}_{\text{KL}}\left( q_{w|x}\|p_{w|y} \right).$$

This objective corresponds to the assumptions $x \perp y \mid z, w$ and $z \perp w$ unconditionally, and therefore does not match the true posterior dependencies once conditioned on $x$. Furthermore, since the objective does not include an adversarial term acting on $z$, it may still encode information regarding $y$.

**Characteristic-capturing VAEs (CCVAE)** (Joy et al., 2020) assume the same probabilistic model as DIVA (shown in Supplementary Figure 1b), but optimize a different objective,

$$\mathbb{E}_{q_{z,w|x,y}}\left[ \frac{q(y \mid w)}{q(y \mid x)} \log \frac{p(x \mid z, w)}{q(y \mid w)} \right]$$
$$- \mathcal{D}_{\text{KL}}\left( q_{z|x}\|p_{z|y} \right) - \mathcal{D}_{\text{KL}}\left( q_{w|x}\|p_{w|y} \right) + \log q(y \mid x).$$

Here, the prior $p_{z|y}$ encodes the assumption that $z$ should depend on $y$, and therefore allows the invariant representation to in fact be condition-specific.

**Summary and comparison:** Prior methods either lack an explicit mechanism forcing the shared latent $z$ to discard label information (Sohn et al., 2015; Ilse et al., 2020), reconstruct $x$ solely from the condition-specific representation, thereby allowing the invariant representation $z$ to remain uninformative (Klys et al., 2018), impose an unconditional independence assumption $z \perp w$ that does not match the true conditional independencies (Ilse et al., 2020), or encode in the prior that the invariant representation $z$ should in fact depend on $y$ (Joy et al., 2020).

Our method addresses these limitations by (i) optimizing a *principled probabilistic objective* that enforces the correct conditional independencies, (ii) placing a *prior over $w$ conditioned on the mean of $z$*, which keeps $z$ informative while discouraging leakage of information through label-specific aggregation, (iii) using *two distinct reconstruction paths*, from the invariant and condition-specific representations, which further compel $z$ to capture informative shared structure, and (iv) incorporating an *adversarial term* that penalizes leakage of class-specific information into the invariant representation $z$.

## 5. Experiments

**Datasets:** We evaluate DisCoVR against existing approaches on synthetic data, natural images, and biological data. These datasets were chosen to probe condition-invariant structure and to ensure comparability with prior work. For instance, Swiss rolls and CelebA were used in Klys et al. (2018), and CelebA also in Joy et al. (2020).

**Evaluation:** When applicable, we evaluate reconstruction quality using negative log-likelihood (NLL), root mean squared error (RMSE), and the absolute deviation from the optimal-Bayes classifier on the reconstructed data, denoted as $\Delta$-Bayes.

Disentanglement is quantified via a neural estimator of the mutual information $I(z; w)$ (Belghazi et al., 2018), and additional disentanglement metrics are reported in Appendix E. Full model architectures, hyperparameters, and additional implementation details are provided in Appendix H.

Our results show that DisCoVR achieves superior performance across all experiments.

### 5.1. Simulated data

We begin with controlled synthetic experiments to isolate and visualize disentanglement.

#### 5.1.1. PARAMETRIC MODEL

**Data generating model:** Consider a model where the observed data $x$ is generated as a function of two latent variables $z$ and $w$, and $y$ is a binary label. Assume that the marginal distributions of the latent variables are given by $z \sim \mathcal{N}(0, 1)$ and $w \sim \mathcal{N}(0, 1)$, and that the data $x$ is generated as the sum of the two latent variables: $x = z + w$. Since $z$ and $w$ are both drawn from $\mathcal{N}(0, 1)$, it follows that $x \sim \mathcal{N}(0, 2)$. Finally, assume that the binary label is determined by the sign of $w$: $y = 1$ if $w > 0$, and $y = 0$ otherwise.

**Optimal disentanglement:** Given that $z$ and $w$ are independent and $x = z + w$, we have that $p(z \mid x) = \mathcal{N}(z; \frac{x}{2}, \frac{1}{2})$. Hence, given $x$, the best estimate for $z$ is $\frac{x}{2}$. Note that when

ignoring the label $y$, the conditional distribution $p(w \mid x)$ is $p(w \mid x) = \mathcal{N}(w; \frac{x}{2}, \frac{1}{2})$. However, the observation of $y$ (which indicates whether $w$ is positive or negative) truncates this distribution:

$$p(w|x, y=1) = \frac{\mathcal{N}(w; \frac{x}{2}, \frac{1}{2})}{\Phi\left(\frac{x}{\sqrt{2}}\right)}, \quad p(w|x, y=0) = \frac{\mathcal{N}(w; \frac{x}{2}, \frac{1}{2})}{1 - \Phi\left(\frac{x}{\sqrt{2}}\right)}$$

**Results:** Table 1 shows that DisCoVR (ours) best matches the analytic posteriors, yielding the lowest Bayes-classifier deviation and reconstruction error.

#### 5.1.2. NOISY SWISS ROLL

**Dataset:** We use a noisy variant of the labeled Swiss Roll dataset (Marsland, 2014; Klys et al., 2018), generating $n = 20,000$ samples and assigning binary labels based on a lengthwise split, with labels flipped at rate $\rho$. The common geometry (its projection along the 2D plane) remains intact, while the conditional structure along the third axis becomes noisy. Figure 4A illustrates the setup.

**Optimal disentanglement:** Since the Swiss Roll is sliced at the center and label noise is applied uniformly, marginalizing over labels yields a symmetric spiral centered along the roll—i.e., the marginal posterior $p(z \mid x)$ is label-invariant. In contrast, the conditional component retains a noisy but informative signal, with a uniform noise level of $\rho = 0.3$. As a result, the Bayes optimal classifier trained on any realistic representation is upper-bounded at 70% accuracy.

**Results:** Figure 4 presents qualitative and quantitative results, showing that DisCoVR both models the label noise accurately and effectively disentangles shared and condition-specific structure. Notably, DisCoVR captures the marginal data distribution, successfully recovering the expected spiral pattern, as shown in Figure 4B.

Additionally, the results in Table 2 show that DisCoVR achieves the lowest deviation from the optimal Bayes classifier and minimal information leakage between latent variables, while preserving reconstruction quality. This confirms that label information is concentrated in $w$ while $z$ remains both informative and label-invariant.

### 5.2. Real data

#### 5.2.1. NOISY COLORED MNIST

**Dataset:** We construct a modified MNIST (Deng, 2012) dataset from $60,000$ duplicated images: in one copy we remove the red channel ($y = 0$) and in the other we remove the green channel ($y = 1$), so that the digit shape remains intact and is carried entirely by the blue channel. Label noise is introduced by flipping $y$ with probability $\rho \in \{0, 0.1, 0.2, 0.3, 0.4\}$.

*Table 1.* Parametric model results: DisCoVR (ours) outperforms all competitors across all metrics.

|  | NLL ↓ | $\mathcal{D}_{\mathrm{KL}}(q_{z|x} \,\|\, p_{z|x}) \downarrow$ | $\mathcal{D}_{\mathrm{KL}}(q_{w|x,y} \,\|\, p_{w|x,y}) \downarrow$ | $\Delta - \mathrm{Bayes} \downarrow$ |
|---|---|---|---|---|
| CSVAE No Adv. | $1.810 \pm 0.016$ | $6.65 \pm 3.46$ | $23.61 \pm 0.36$ | $24.83 \pm 0.04$ |
| CSVAE | $1.786 \pm 0.022$ | $2.85 \pm 1.11$ | $23.98 \pm 4.36$ | $24.33 \pm 1.28$ |
| HCSVAE No Adv. | $1.784 \pm 0.010$ | $4.01 \pm 0.07$ | $25.82 \pm 0.38$ | $24.99 \pm 0.01$ |
| HCSVAE | $1.770 \pm 0.004$ | $3.99 \pm 0.09$ | $26.25 \pm 0.59$ | $24.99 \pm 0.01$ |
| DIVA | $1.788 \pm 0.008$ | $3.21 \pm 1.52$ | $12.88 \pm 3.31$ | $3.51 \pm 0.32$ |
| CCVAE | $1.785 \pm 0.006$ | $1.77 \pm 0.81$ | $12.95 \pm 3.35$ | $3.57 \pm 0.15$ |
| DisCoVR (ours) | $\mathbf{1.769 \pm 0.003}$ | $\mathbf{0.17 \pm 0.01}$ | $\mathbf{10.10 \pm 0.73}$ | $\mathbf{0.1 \pm 0.28}$ |

*Table 2.* Noisy Swiss roll results: DisCoVR (ours) yields lowest deviation from optimal-Bayes, maintains low latent leakage, and high reconstruction accuracy.

|  | $I(z; w) \downarrow$ | NLL ↓ | $\Delta - \mathrm{Bayes} \downarrow$ |
|---|---|---|---|
| CSVAE No Adv. | $0.047 \pm 0.023$ | $3.303 \pm 0.003$ | $23.88 \pm 12.02$ |
| CSVAE | $0.031 \pm 0.025$ | $\mathbf{3.302 \pm 0.003}$ | $17.99 \pm 14.58$ |
| HCSVAE No Adv. | $0.024 \pm 0.012$ | $\mathbf{3.302 \pm 0.002}$ | $30.00 \pm 0.00$ |
| HCSVAE | $\mathbf{0.002 \pm 0.001}$ | $\mathbf{3.302 \pm 0.002}$ | $30.00 \pm 0.00$ |
| DIVA | $0.336 \pm 0.083$ | $\mathbf{3.302 \pm 0.003}$ | $1.88 \pm 1.05$ |
| CCVAE | $0.502 \pm 0.089$ | $\mathbf{3.302 \pm 0.002}$ | $2.21 \pm 0.84$ |
| DisCoVR (ours) | $0.005 \pm 0.002$ | $\mathbf{3.302 \pm 0.002}$ | $\mathbf{1.14 \pm 0.21}$ |

**Optimal disentanglement:** With colors balanced across labels, the ideal $z$-reconstruction averages colors over labels, retaining one mixed color (Figure 3).

**Results:** We evaluate marginal coloring reconstruction by DisCoVR and previous methods. Under no label noise ($\rho = 0$), all methods perform similarly (see Supplementary Figure 3).

However, at all non-zero noise levels, DisCoVR consistently outperforms competing methods and is the only approach that reconstructs digits in purple, correctly averaging over the two colors.

Metrics for $\rho = 0.3$ are shown in Supplementary Table 2, with results for other noise levels in Supplementary Figure 4.

### 5.2.2. CELEBA

**Glasses attribute:** We use all CelebA (Liu et al., 2015) images labeled with *eyeglasses* attribute ($y = 1$), and twice as many randomly sampled images without glasses ($y = 0$), resulting in $n = 35,712$ images in total.

**Hat attribute:** Results for an analogous experiment with the wearing-hat attribute are provided in Appendix G.

**Results:** Figure 5 shows that DisCoVR accurately reconstructs input images while producing shared embeddings that marginalize over the *eyeglasses* attribute, consistently adding "pseudo-glasses" to all samples.

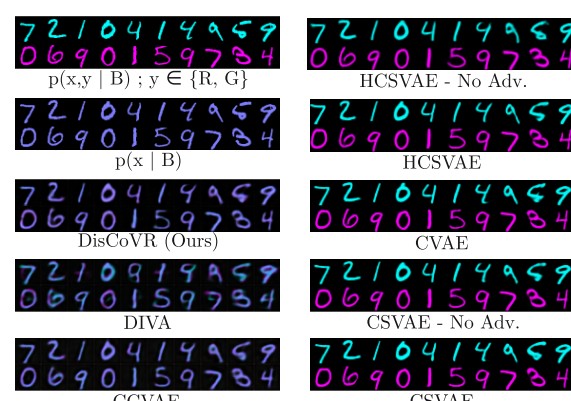

*Figure 3.* Colored MNIST reconstructions from the label-agnostic representation $z$ at noise level $\rho = 0.3$. Only DisCoVR consistently produces mixed semi-red/blue (purple) digits, indicating that color information has been removed from $z$ and that the reconstructions approximate the true marginal.

Competing methods are shown in Supplementary Figure 5, with quantitative results in Supplementary Table 3.

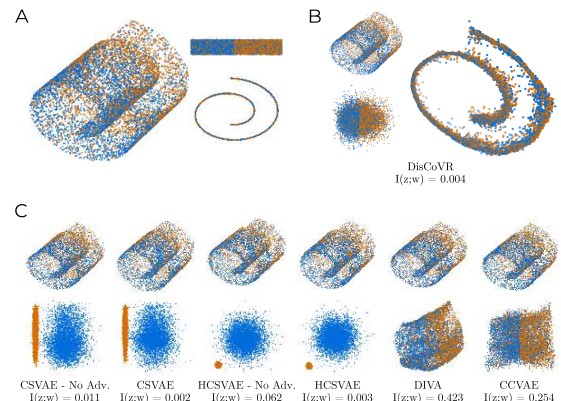

*Figure 4.* A: Noisy labeled Swiss Roll dataset. B: DisCoVR recovers the conditional embedding and reconstruction (left), while the shared embedding recovers the marginal spiral structure (right). C: Across models, ours best matches the disentangled ideal: $z$ captures the clean Swiss-roll geometry independently of the label, while label variation is isolated in $w$.

While full reconstruction quality from both representations together is comparable across methods, DisCoVR achieves notably better disentanglement. In this experiment, however, the adversarial classifier incurs higher computational cost compared to other methods. See Table 15 for additional details.

A

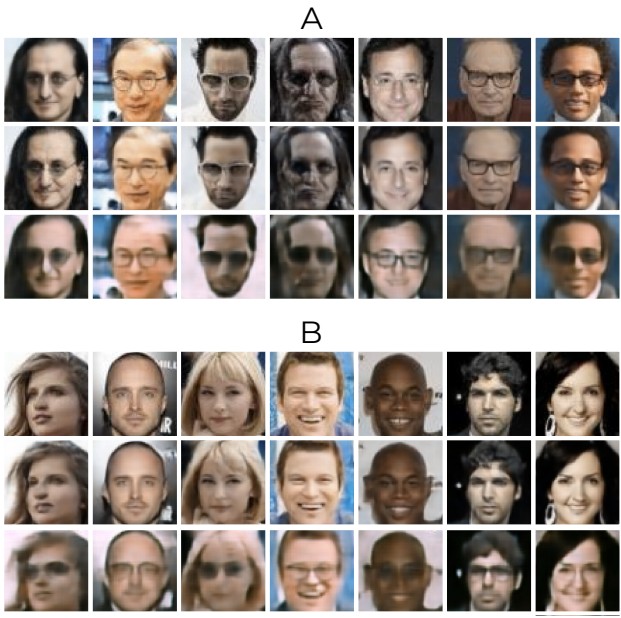

B

*Figure 5.* CelebA-Eyeglasses results. Top: Original images with (A) or without eyeglasses (B). Middle: Full reconstructions by DisCoVR. Bottom: reconstructions solely from invariant embeddings $z$. The condition-specific representation needs to be invariant to $y$ (presence or absence of glasses). Indeed, all reconstructed faces display an intermediate "pseudo-glasses" appearance in both A and B, regardless of their presence in the original images.

### 5.2.3. SINGLE CELL RNA-SEQUENCING

**Dataset:** We analyze single-cell RNA sequencing from $n = 13,999$ peripheral blood mononuclear cells (PBMCs) collected from 8 lupus patients under two conditions: 7,451 cells control ($y = 0$), and 6,548 IFN-$\beta$ stimulation cells. IFN-$\beta$ stimulation induces notable shifts in gene expression, visible in the UMAP embedding in Figure 6B (left).

**Results:** Supplementary Table 19 shows that DisCoVR effectively achieves the desired behavior with strong empirical performance, where only cell type information is captured in $z$ (Figure 6A, middle) while the effects of IFN-$\beta$ stimulation are wholly represented in $w$ (Figure 6B, right). Other approaches either (1) achieve mixing in the $z$ space, but compromise on keeping cell types separated or (2) leak information about stimulation into the $z$ space (Supplementary Figure 6).

**Facilitating interpretability:** By enabling marginalized reconstructions, DisCoVR provides a direct link be-

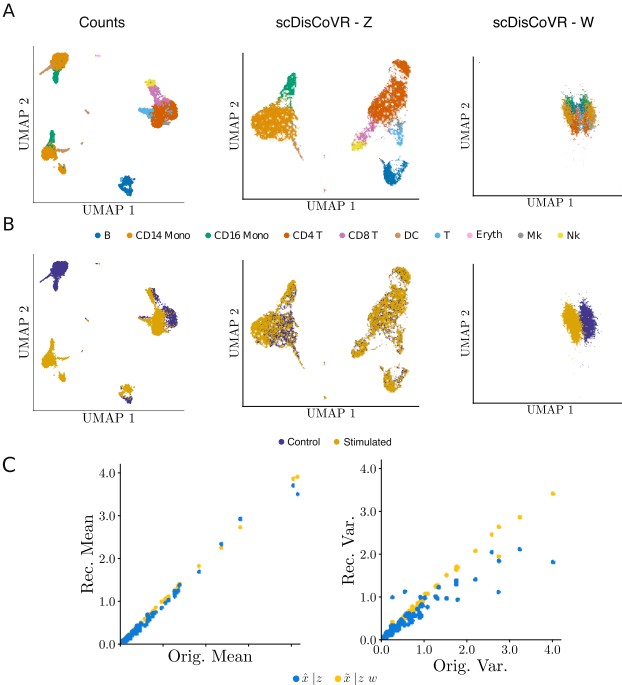

*Figure 6.* A–B (left): UMAPs of raw gene counts from the IFN-$\beta$ dataset. A–B (middle): Shared embedding $z$ aligns cells by type while removing stimulation effects. A–B (right): Condition-specific embedding $w$ isolates the stimulation effect. C: Reconstructions from both $z$ and $w$ (yellow) recover empirical gene means and variances, while reconstructions from $z$ alone (blue) miss the stimulation-induced variance, confirming that $z$ discards $y$ while preserving cell-type features.

tween shared embeddings and gene expression, offering clearer insight into the effects of IFN-$\beta$ stimulation, unlike other methods. In Figure 6C, comparing variance across marginal and full reconstructions accurately recovers gene-level differences associated with IFN-$\beta$ stimulation, including *ISG15, FTL, CCL8, CXCL10, CXCL11, APOBEC3A, IL1RN, IFITM3* and *RSAD2*.

## 6. Conclusion

In this work we introduced a variational framework for disentangled representation learning in multi-condition datasets that explicitly separates condition-invariant and condition-specific factors. Unlike prior work, DisCoVR is built around a principled probabilistic objective that encodes the correct conditional independencies, a prior over $w$ conditioned on the class-wise mean of $z$, and an adversarial term that limits label information in the invariant representation. The model uses two reconstruction paths, which forces $z$ to remain informative about shared structure.

Across synthetic benchmarks and real-world datasets, DisCoVR achieves strong reconstruction, low information leakage, and accurate modeling of conditional effects, con-

sistently outperforming existing methods in disentangling shared and condition-specific structure.

## Acknowledgements

BD acknowledges the support of the CIFAR MacMillan Multiscale Human Project and the National Institute of General Medical Sciences of the National Institutes of Health under award number R35 GM157082-02. DB acknowledges the support of NSF IIS-2127869, NSF DMS-2311108, ONR N000142412243, and the Simons Foundation. YS acknowledges the support of the Founder's Postdoctoral Fellowship, Department of Statistics, Columbia University.

## Impact Statement

This paper presents work whose goal is to advance the field of Machine Learning. There are many potential societal consequences of our work, none which we feel must be specifically highlighted here.

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

# A. Additional related work

## A.1. Domain generalization

The task of representation disentanglement is closely related to the field of domain generalization (Muandet et al., 2013), which assumes limited or no access to target domain samples and aims to learn representations that can be readily adapted, often via transfer learning, to new, unseen domains.

As noted by Wang et al. (2019), existing methods in domain generalization can be broadly categorized into two main approaches: (i) approaches for reducing the inter-domain differences, often by using adversarial techniques (Ghifary et al., 2015; Wang et al., 2017; Motiian et al., 2017; Li et al., 2018; Carlucci et al., 2019; Wang et al., 2019; Akuzawa et al., 2020; Zhu et al., 2022; Gokhale et al., 2023; Dayal et al., 2023; Cheng et al., 2023; Chen et al., 2024), and (ii) Approaches that construct an ensemble of domain-specific models, and then fuse their representations to form a unified, domain-agnostic representation (Ding & Fu, 2017; Mancini et al., 2018; Zhou et al., 2021; Muhammad et al., 2024).

Additional strategies for domain generalization include contrastive learning approaches (Kim et al., 2021), methods based on distribution alignment via metrics (Muandet et al., 2013; Sun & Saenko, 2016), and techniques utilizing custom network architectures, for instance by incorporating domain-specific adapters between shared layers (Rebuffi et al., 2017; 2018; Li & Vasconcelos, 2019; Kazuki OMI, 2022).

The primary distinction between these methods and ours lies in the explicit probabilistic modeling and disentanglement of domain-invariant and domain-specific factors. Whereas prior approaches typically focus on aligning domains through adversarial training or fusing multiple domain-specific predictors, our method constructs a structured latent space, decomposed into a condition-specific representation $z$, capturing domain-invariant information, and a conditional component $w$, which encodes domain-specific variability. This factorization is learned through a tailored variational objective involving an adversarial penalty and two reconstructions —one based on $z$ alone, and another on the full latent pair $(z, w)$, thereby promoting both interpretability and a clean separation of shared and domain-aware features.

## A.2. Out of distribution generalization

### A.2.1. ENVIRONMENT BALANCING METHODS

The field of out-of-distribution (OOD) generalization emerged from foundational work on causality and invariance across training environments (Peters et al., 2016; 2017). The central assumption is that each environment exhibits distinct spurious correlations between features and labels; therefore, robust generalization requires models to focus on invariant relationships that hold across environments. To address this distribution shift, many recent approaches adopt a regularized empirical risk minimization framework:

$$\min_{\theta} \sum_{e \in E_{\text{train}}} \ell^e(f_\theta) + \lambda R(f_\theta, E_{\text{train}}), \tag{10}$$

where the regularizer $R$ encourages representations that are stable across environments. Among these, Invariant Risk Minimization (IRM) (Arjovsky et al., 2019) enforces that a single classifier remains optimal across all environments, Variance Risk Extrapolation (VarREx) (Krueger et al., 2021) promotes robustness by minimizing the variance of losses across environments, and CLOvE (Wald et al., 2021) takes a calibration-theoretic perspective, penalizing discrepancies between predicted confidence and correctness across environments.

While these methods focus on enforcing predictive invariance across environments through regularization, our approach instead explicitly enforces conditional independence between the shared latent variable $z$ and an environment-aware variable $w$.

### A.2.2. DISTRIBUTIONALLY ROBUST METHODS

An alternative line of work for handling distribution shifts is Distributionally Robust Optimization (DRO) (Ben-Tal et al., 2013; Duchi et al., 2021; Duchi & Namkoong, 2021; Wei et al., 2023), which avoids assuming a fixed data-generating distribution. Instead, DRO methods optimize performance under the worst-case scenario over a family of plausible distributions. A prominent variant, known as group DRO (Sagawa et al., 2019; Piratla et al., 2021), introduces group-level structure that may correlate with spurious features, potentially leading to biased predictions. In settings where group labels are not directly observed, several strategies have been proposed, including reweighting high-loss examples (Liu et al., 2021) and balancing class-group combinations through data sub-sampling (Idrissi et al., 2022).

However, these approaches assume that the label space remains fixed between training and test time, limiting their applicability in adaptation to new domains, environments or conditions.

### A.3. Zero-shot learning

Zero-shot learning systems (Fei-Fei et al., 2006; Larochelle et al., 2008) aim to classify instances from novel, previously unseen classes at test time. In contrast to the out-of-distribution (OOD) generalization setting, these approaches typically do not assume the presence or structure of a distribution shift. Instead, a common strategy is to learn data representations that capture class-agnostic similarity, enabling the model to determine whether two instances belong to the same class without requiring knowledge of the class identity itself. Such methods include contrastive-learning (Hadsell et al., 2006), siamese neural networks (Koch et al., 2015), triplet networks (Hoffer & Ailon, 2015), and other more recent variations (Oh Song et al., 2016; Sohn, 2016; Wu et al., 2017; Yuan et al., 2019). Recent work has begun to address the impact of class distribution shifts in zero-shot settings. For instance, Slavutsky & Benjamini (2024) integrate environment-based regularization—motivated by OOD generalization—with zero-shot learning by simulating distribution shifts through hierarchical sampling, enabling the model to learn representations that are robust to shifts in class distributions.

While this line of work shares our motivation of improving robustness under unseen conditions, it primarily addresses the problem of class-level generalization through similarity-based learning, rather than explicitly modeling and disentangling the latent factors—such as domain or environment—that drive distributional variation across tasks.

## B. Proofs

### B.1. Proof of Proposition 2.1

*Proof.*

$$\text{ELBO}(q, p; x, y) - \mathcal{L}_w(q_{w|x,y}, p; x, y) \tag{11}$$

$$= \left[\log p(x \mid y) - \mathcal{D}_{\text{KL}}(q_{w|x,y} \,\|\, p_{w|x,y})\right] - \left[\log p(x \mid y) - \mathcal{D}_{\text{KL}}\left(q_{z|x}q_{w|x,y} \,\|\, p_{z,w|x,y}\right)\right] \tag{12}$$

$$= \mathcal{D}_{\text{KL}}\left(q_{z|x}q_{w|x,y} \,\|\, p_{z,w|x,y}\right) - \mathcal{D}_{\text{KL}}\left(q_{w|x,y} \,\|\, p_{w|x,y}\right) \tag{13}$$

$$= \mathbb{E}_{q_{w|x,y}}\left[\mathbb{E}_{q_{z|x}}\left[\log q(z \mid x) + \log q(w|x,y) - \log p(z, w \mid x, y)\right]\right] \tag{14}$$

$$\quad - \mathbb{E}_{q_{w|x,y}}\left[\log q(w|x,y) - \log p(w|x,y)\right] \tag{15}$$

$$= \mathbb{E}_{q_{w|x,y}}\left[\mathbb{E}_{q_{z|x}}\left[\log q(z \mid x) - \log p(z, w \mid x, y) + \log p(w|x,y)\right]\right] \tag{16}$$

$$= \mathbb{E}_{q_{w|x,y}}\left[\mathbb{E}_{q_{z|x}}\left[\log q(z \mid x) - \log p(z \mid w, x, y)\right]\right] \tag{17}$$

$$= \mathbb{E}_{q_{w|x,y}}\left[\text{KL}\left(q_{z|x} \,\|\, p_{z|w,x,y}\right)\right]. \tag{18}$$

$\square$

### B.2. Game equilibrium

#### B.2.1. REGULARITY CONDITIONS

To ensure that expectations and KL-terms in the game objective $\mathcal{L}(q_{z|x}, q_{w|x,y}, g_{y|z})$ render the functionals strictly concave in $q_{z|x}$, strictly concave in $q_{w|x,y}$, and strictly convex in $g$, the following regularity conditions are required:

1. The likelihoods $p(x|z), p(x|z, w), p(y|x)$ are strictly positive, continuous densities.

2. The variational families $Q_z$ and $Q_w$, and the set of achievable classifiers $\mathcal{G}$ are non-empty, convex and compact.

3. $\log p(x|z, w)$ and $\log g(y|x)$ are integrable.

#### B.2.2. PROOF OF PROPOSITION 2.2

*Proof.* Since $\mathcal{L}_z(q_{z|x}, p; x)$ is the standard ELBO objective, we have that

$$\mathcal{L}_z(q_{z|x}, p; x) = \log p(x) - \mathcal{D}_{\text{KL}}\left(q_{z|x} \,\|\, p_{z|x}\right). \tag{19}$$

Similarly, we have that

$$\mathcal{L}_w(q_{w|x,y}, p; x, y) = \log p(x \mid y) - \mathcal{D}_{\mathrm{KL}}\left(q_{z|x} q_{w|x,y} \,\|\, p_{z,w|x,y}\right). \tag{20}$$

Thus,

$$\mathcal{L}(q_{z|x}, q_{w|x,y}, g_{y|z}) = \mathbb{E}_{p_{x,y}}\Big[\log p(x) - \mathcal{D}_{\mathrm{KL}}\left(q_{z|x} \,\|\, p_{z|x}\right) \tag{21}$$

$$+ \log p(x \mid y) - \mathcal{D}_{\mathrm{KL}}\left(q_{z|x} q_{w|x,y} \,\|\, p_{z,w|x,y}\right) \tag{22}$$

$$- \mathbb{E}_{q_{z|x}} \log g(y \mid z)\Big]. \tag{23}$$

For fixed $q_{z|x}$, the adversarial classifier minimizes:

$$-\mathbb{E}_{p_{x,y}} \mathbb{E}_{q_{z|x}} \log g(y \mid z), \tag{24}$$

which is the population cross-entropy and is strictly convex in $g(y|z)$, and thus has a unique solution.

It remains to show that the terms in the objective function that depend on $q_{z|x}$ and $q_{w|x,y}$, are strictly concave in each argument when the others are held fixed.

Focusing on the terms dependent on $q_{w|x,y}$ first, define

$$\ell_w := -\,\mathcal{D}_{\mathrm{KL}}\left(q_{z|x} q_{w|x,y} \,\|\, p_{z,w|x,y}\right) \tag{25}$$

$$= -\iint q(z \mid x)\, q(w \mid x, y)\left[\log q(z \mid x) + \log q(w \mid x, y) - \log p(z, w \mid x, y)\right] dz\, dw$$

$$= -\int q(z \mid x) \log q(z \mid x)\, dz - \int q(w \mid x, y) \log q(w \mid x, y)\, dw \tag{26}$$

$$+ \iint q(z \mid x)\, q(w \mid x, y) \log p(z, w \mid x, y)\, dz\, dw \tag{27}$$

$$= H(q_{z|x}) + H(q_{w|x,y}) + \mathbb{E}_{q_{z|x}} \mathbb{E}_{q_{w|x,y}} \log p(z, w \mid x, y). \tag{28}$$

Note that

$$\mathbb{E}_{q_{z|x}} \mathbb{E}_{q_{w|x,y}} \log p(z, w \mid x, y) \tag{29}$$

is linear in $q_{w|x,y}$, and since $H(q_{w|x,y})$ is strictly concave in $q_{w|x,y}$, we have that $\mathbb{E}_{p_{x,y}}[\ell_w]$ is strictly concave in $q_{w|x,y}$.

Similarly, define

$$\ell_z := -\mathcal{D}_{\mathrm{KL}}\left(q_{z|x} \,\|\, p_{z|x}\right) - \mathcal{D}_{\mathrm{KL}}\left(q_{z|x} q_{w|x,y} \,\|\, p_{z,w|x,y}\right). \tag{30}$$

By convexity of KL divergence in its first argument, $-\mathcal{D}_{\mathrm{KL}}\left(q_{z|x} \,\|\, p_{z|x}\right)$ is strictly concave in $q_{z|x}$.

Focusing on the second KL term, from Equation 28 we have that

$$-\mathcal{D}_{\mathrm{KL}}\left(q_{z|x} q_{w|x,y} \,\|\, p_{z,w|x,y}\right) = H(q_{z|x}) + H(q_{w|x,y}) + \mathbb{E}_{q_{z|x}} \mathbb{E}_{q_{w|x,y}} \log p(z, w \mid x, y), \tag{31}$$

where $H(q_{z|x})$ is strictly concave in $q_{z|x}$.

Recall that we assumed that $p(w|y)$ depends on $q(z|x)$. Under our model

$$p(x, y, z, w) = p(y)p(w \mid y)p(z)p(x \mid z, w), \tag{32}$$

yielding

$$p(z, w \mid x, y) = p(w \mid y)\frac{p(z)p(x \mid z, w)}{p(x \mid y)}. \tag{33}$$

Hence,

$$\mathbb{E}_{q_{z|x}} \mathbb{E}_{q_{w|x,y}} \log p(z, w \mid x, y) = \mathbb{E}_{q_{z|x}} \mathbb{E}_{q_{w|x,y}}\left[\log p(w \mid y) + \log \frac{p(z)p(x \mid z, w)}{p(x \mid y)}\right],$$

where $p(w \mid y) = \mathcal{N}(w; \mu_y, I)$ with $\mu_y = \mathbb{E}_{p_{x|y}}\left[\mathbb{E}_{q_{z|x}}[z]\right]$. Therefore,

$$\mathbb{E}_{q_{z|x}}\mathbb{E}_{q_{w|x,y}}\left[\log p(w \mid y)\right] = -\frac{1}{2}\left[d\log(2\pi) + \mathbb{E}_{q_{z|x}}\mathbb{E}_{q_{w|x,y}}\|w - \mu_y\|^2\right] \tag{34}$$

where $-\|w - \mu_y\|^2$ is a quadratic form in $\mu_y$, which is linear in $q_{z|x}$, and thus $\mathbb{E}_{q_{z|x}}\mathbb{E}_{q_{w|x,y}}\left[\log p(w \mid y)\right]$ is strictly concave in $q_{z|x}$. Hence, $-\mathcal{D}_{\mathrm{KL}}\left(q_{z|x}q_{w|x,y} \| p_{z,w|x,y}\right)$ is strictly concave in $q_{z|x}$, and thus so is $\mathbb{E}_{p_{x,y}}[\ell_z]$. $\qquad\square$

# C. Supplementary Figures

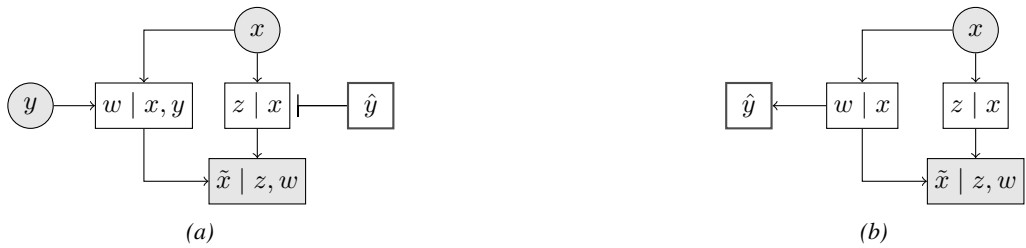

*(a)*             *(b)*

*Figure 1.* Encoder-decoder structures for previous approaches. (a) CSVAE. (b) DIVA - CCVAE.

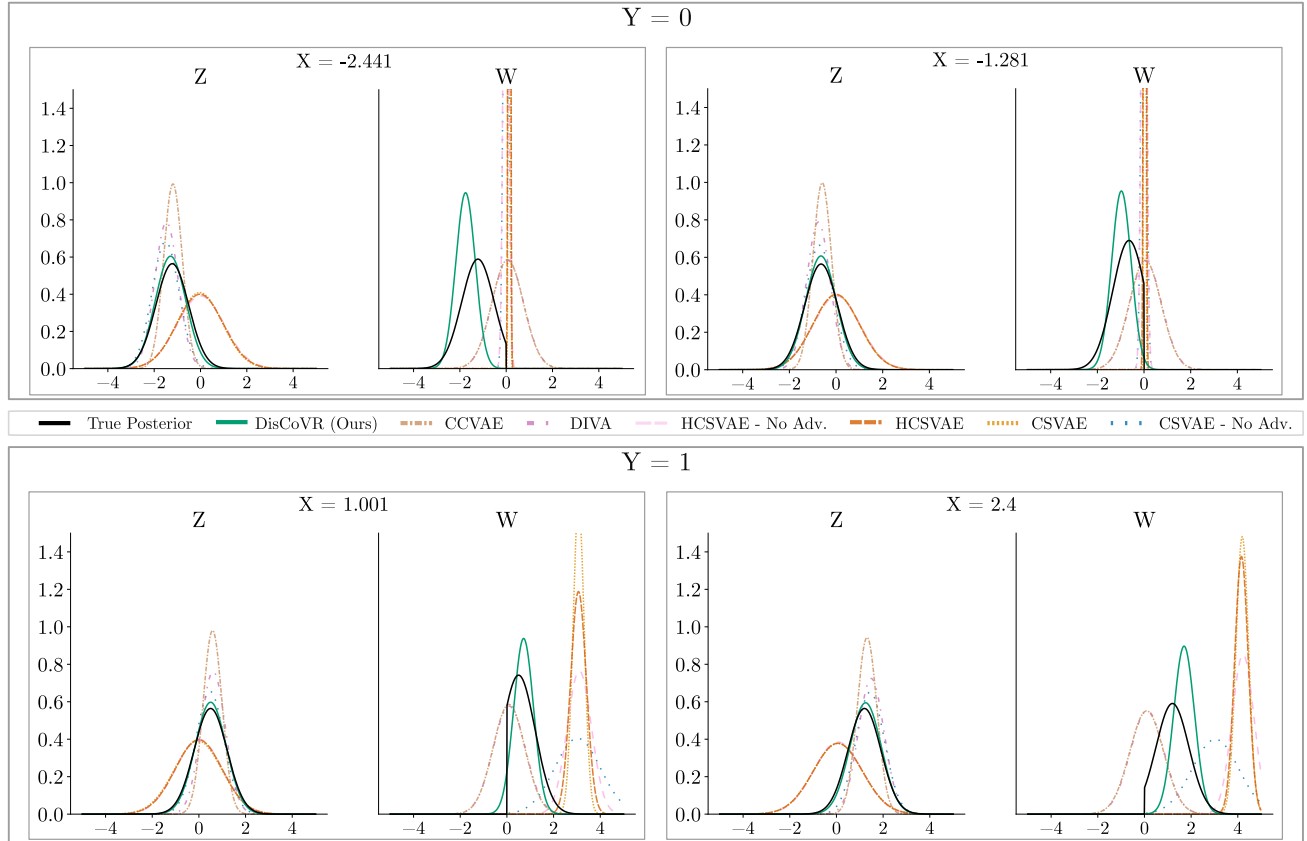

*Figure 2.* Comparison of approximate variational posteriors against the true posterior for latent variables $z, w$ for different values of $x$ with $y = 0$ (top) and $y = 1$ (bottom).

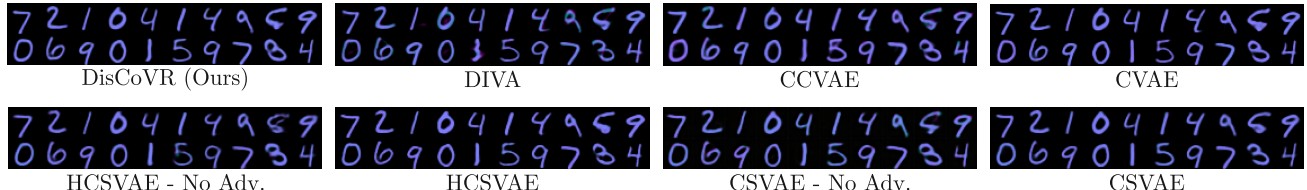

*Figure 3.* Colored MNIST results for no noise.

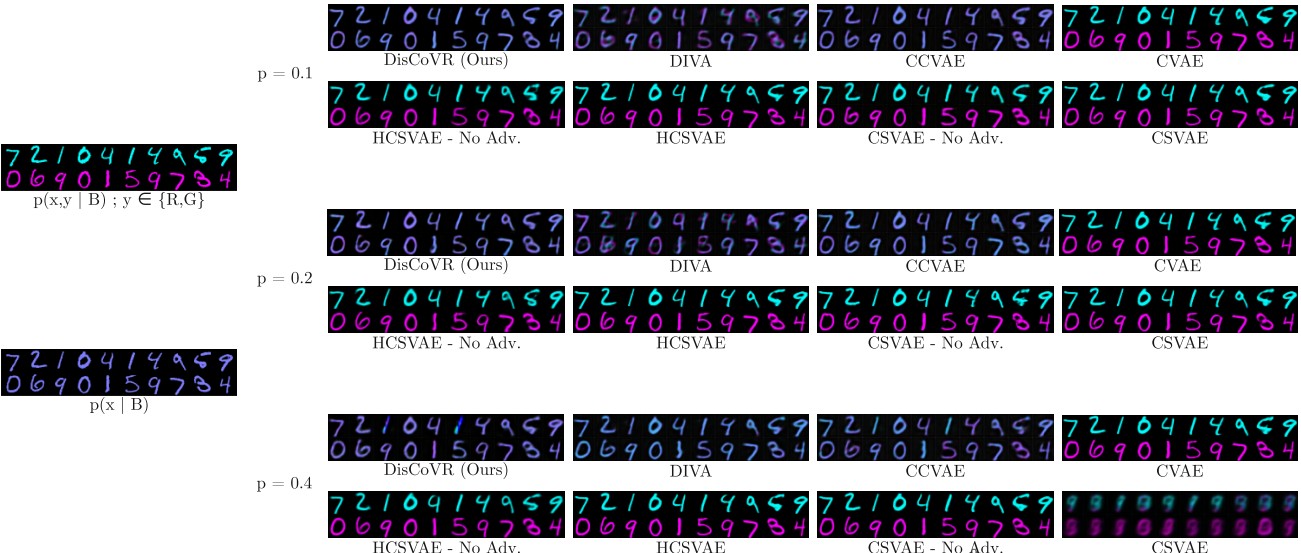

*Figure 4.* Colored MNIST visual results across the remaining noise levels.

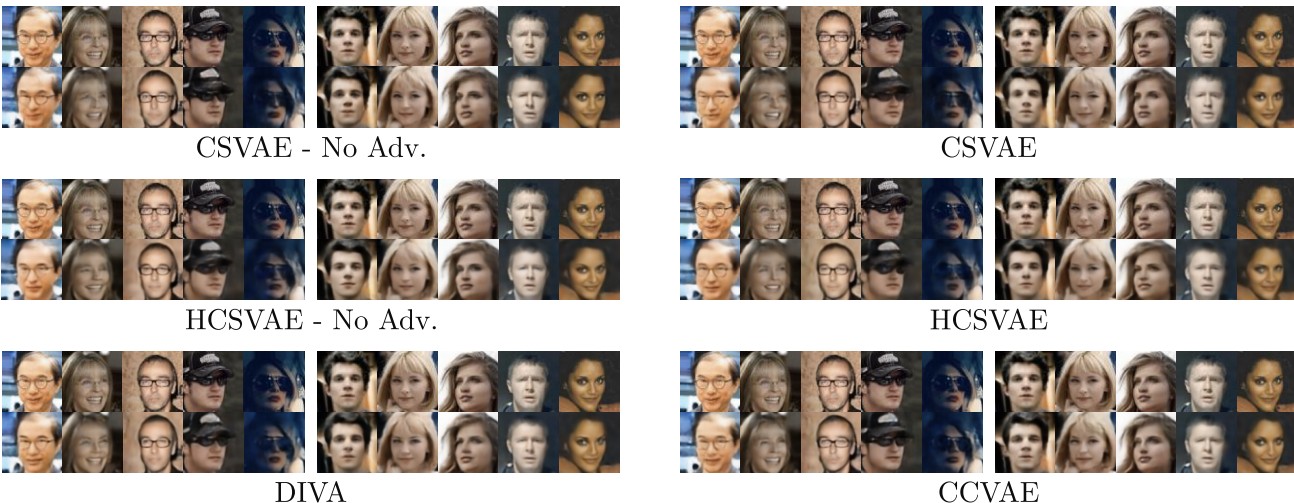

*Figure 5.* Reconstruction performance for other models on the CelebA-Glasses dataset. Top: Original samples from the data. Bottom: Reconstructions by the given model.

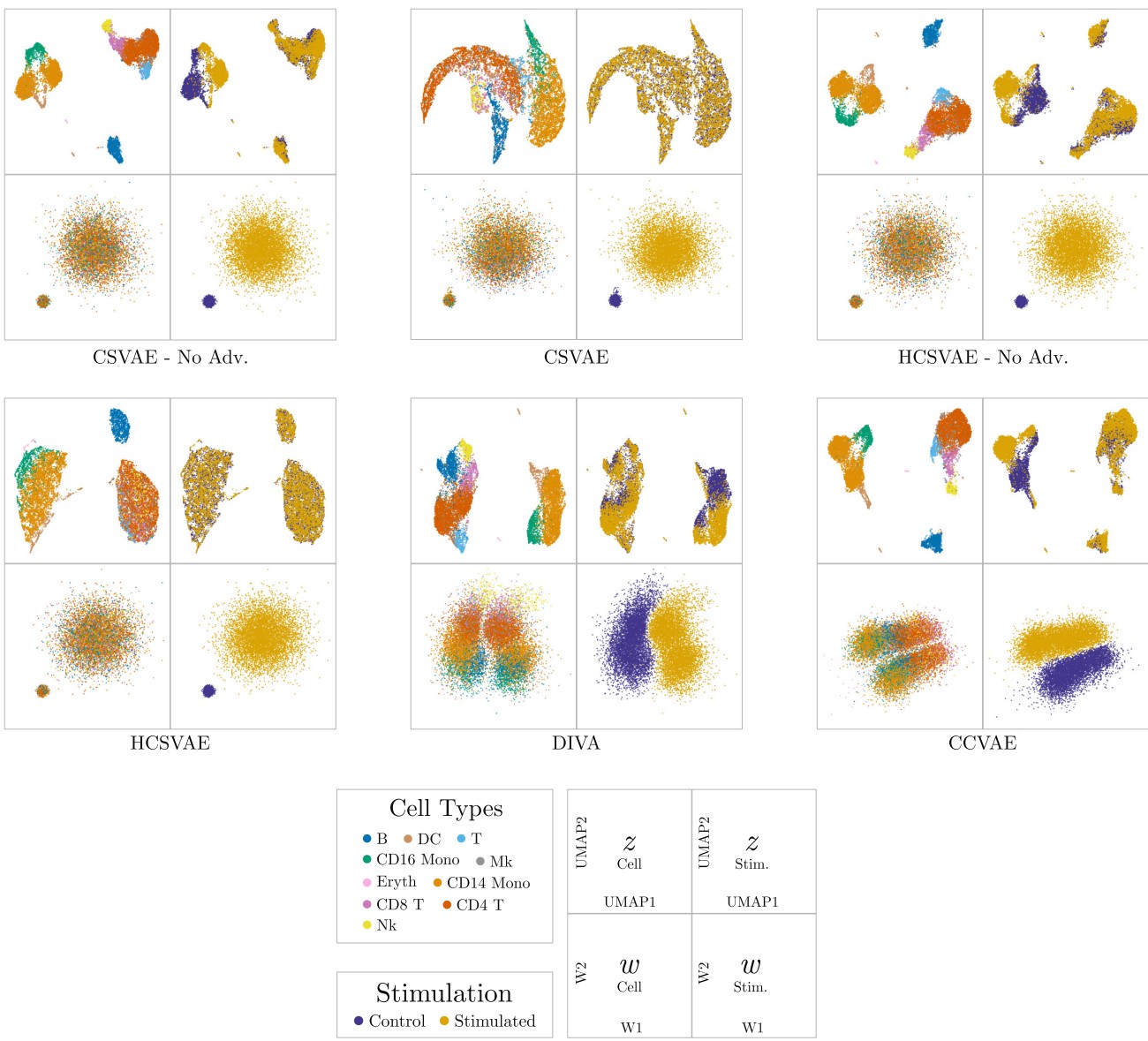

*Figure 6.* Embeddings obtained by other models on the Kang dataset. For each block, top (resp. bottom) rows are $z$ (resp. $w$) embeddings, while left (resp. right) columns are colored by cell type (resp. stimulation).

# D. Supplementary tables for experimental results

*Table 1.* RMSE for the Colored MNIST dataset without any label noise.

|  | Marginal RMSE ($p = 0$) ↓ |
|---|---|
| CSVAE - No Adv. | $\mathbf{0.064 \pm 0.002}$ |
| CSVAE | $0.079 \pm 0.008$ |
| HCSVAE - No Adv. | $0.094 \pm 0.004$ |
| HCSVAE | $0.079 \pm 0.030$ |
| DIVA | $0.065 \pm 0.005$ |
| CCVAE | $0.065 \pm 0.006$ |
| DisCoVR (ours) | $\mathbf{0.064 \pm 0.000}$ |

*Table 2.* RMSE calculated between the estimated and true marginal across different levels of label noise on the Colored MNIST dataset. $p$ defines label flip probability. Bold denotes best performance.

|  | Marginal RMSE ↓ | | | |
|---|---|---|---|---|
|  | $p = 0.1$ | $p = 0.2$ | $p = 0.3$ | $p = 0.4$ |
| CSVAE - No Adv. | $0.141 \pm 0.002$ | $0.141 \pm 0.003$ | $0.142 \pm 0.002$ | $0.143 \pm 0.002$ |
| CSVAE | $0.135 \pm 0.022$ | $0.152 \pm 0.018$ | $0.181 \pm 0.007$ | $0.173 \pm 0.008$ |
| HCSVAE - No Adv. | $0.150 \pm 0.001$ | $0.150 \pm 0.000$ | $0.151 \pm 0.000$ | $0.151 \pm 0.001$ |
| HCSVAE | $0.139 \pm 0.003$ | $0.141 \pm 0.001$ | $0.141 \pm 0.001$ | $0.141 \pm 0.001$ |
| DIVA | $0.115 \pm 0.011$ | $0.102 \pm 0.013$ | $0.106 \pm 0.010$ | $0.113 \pm 0.014$ |
| CCVAE | $0.092 \pm 0.002$ | $0.103 \pm 0.014$ | $0.099 \pm 0.011$ | $0.092 \pm 0.005$ |
| DisCoVR (ours) | $\mathbf{0.073 \pm 0.001}$ | $\mathbf{0.083 \pm 0.004}$ | $\mathbf{0.087 \pm 0.002}$ | $\mathbf{0.087 \pm 0.001}$ |

*Table 3.* Model performances on the CelebA-Glasses dataset. Bold denotes best performance.

|  | $I(z; w)$ ↓ | NLL (↓) |
|---|---|---|
| CSVAE - No Adv. | $0.048 \pm 0.014$ | $137.522 \pm 0.155$ |
| CSVAE | $0.079 \pm 0.029$ | $145.989 \pm 0.336$ |
| HCSVAE - No Adv. | $0.055 \pm 0.012$ | $131.813 \pm 0.21$ |
| HCSVAE | $0.055 \pm 0.014$ | $137.319 \pm 0.265$ |
| DIVA | $0.188 \pm 0.028$ | $143.528 \pm 0.02$ |
| CCVAE | $0.083 \pm 0.022$ | $\mathbf{131.764 \pm 0.006}$ |
| DisCoVR (ours) | $\mathbf{0.030 \pm 0.011}$ | $135.677 \pm 0.007$ |
| DisCoVR - Common (ours) | — | $374.114 \pm 0.05$ |

# E. Additional disentanglement metrics

We provide an extended disentanglement assessment using multiple metrics. Because mutual information is difficult to estimate reliably, we report two estimators—MINE and kNN. Although their absolute values differ, the relative rankings of the methods remain consistent as can be seen in the ranking tables. In addition to these mutual-information estimates, we also report the following metrics, which quantify the level of label information captured by $w$ compared to $z$ :

**Mutual Information Gap (MIG)**

$$\mathrm{MIG}(w; z) = \frac{I(y; w) - I(y; z)}{H(y)}$$

**Mutual Information Completeness (MIC)**

$$\text{MIC}(w; z) = \frac{I(y; w)}{I(y; w) + I(y; z)}$$

### E.1. Parametric model

CSVAE and its variants impose a fully separable prior, thereby forcing separability even when the true latent structure is not separable (see Table 1). In contrast, DisCoVR learns informative conditional embeddings that closely track the true posterior without requiring ground-truth knowledge of a truncated or fully separable prior, and it outperforms both DIVA and CCVAE.

Replacing the prior in DisCoVR with a fully separable predefined prior on $w$ yields consistent embeddings with the ground-truth structure while retaining the benefits of separability.

*Table 4.* Additional disentanglement metrics calculated with kNN mutual information estimation for the parametric model dataset with $k = 20$. Bold indicates closest to true posterior within group.

| Assumption | Model | $I(y;z)$ | $I(y;w)$ | $I(w;z)$ | $\text{MIG}(w;z)$ | $\text{MIC}(w;z)$ | $I(w;z\mid y)$ |
|---|---|---|---|---|---|---|---|
| | CSVAE - No Adv. | $\mathbf{0.069 \pm 0.034}$ | $0.634 \pm 0.002$ | $\mathbf{0.098 \pm 0.034}$ | $0.063 \pm 0.004$ | $\mathbf{0.904 \pm 0.048}$ | $0.000 \pm 0.002$ |
| | CSVAE | $0.024 \pm 0.048$ | $0.620 \pm 0.044$ | $0.047 \pm 0.050$ | $\mathbf{0.067 \pm 0.010}$ | $0.963 \pm 0.073$ | $0.013 \pm 0.009$ |
| Fully Separable | HCSVAE - No Adv. | $0.000 \pm 0.000$ | $\mathbf{0.643 \pm 0.000}$ | $0.000 \pm 0.001$ | $0.072 \pm 0.000$ | $1.000 \pm 0.000$ | $0.000 \pm 0.000$ |
| | HCSVAE | $0.000 \pm 0.000$ | $\mathbf{0.643 \pm 0.001}$ | $0.001 \pm 0.001$ | $0.072 \pm 0.000$ | $1.000 \pm 0.000$ | $0.000 \pm 0.001$ |
| | DisCoVR (CSVAE prior) | $0.000 \pm 0.000$ | $\mathbf{0.643 \pm 0.000}$ | $0.051 \pm 0.007$ | $0.072 \pm 0.000$ | $1.000 \pm 0.000$ | $\mathbf{0.031 \pm 0.005}$ |
| | DIVA | $0.021 \pm 0.042$ | $0.091 \pm 0.046$ | $0.000 \pm 0.000$ | $0.008 \pm 0.010$ | $0.800 \pm 0.400$ | $0.000 \pm 0.000$ |
| Flexible | CCVAE | $\mathbf{0.022 \pm 0.043}$ | $0.090 \pm 0.045$ | $0.000 \pm 0.000$ | $0.008 \pm 0.010$ | $0.800 \pm 0.400$ | $0.000 \pm 0.000$ |
| | DisCoVR (our prior) | $0.010 \pm 0.006$ | $\mathbf{0.151 \pm 0.007}$ | $\mathbf{0.108 \pm 0.029}$ | $\mathbf{0.016 \pm 0.001}$ | $\mathbf{0.938 \pm 0.035}$ | $\mathbf{0.072 \pm 0.020}$ |
| Fully Separable | Posterior (no truncation) | $0.057 \pm 0.001$ | $0.057 \pm 0.000$ | $0.144 \pm 0.003$ | $0.000 \pm 0.000$ | $0.499 \pm 0.006$ | $0.090 \pm 0.002$ |
| | True Posterior | $0.058 \pm 0.003$ | $0.643 \pm 0.000$ | $0.144 \pm 0.005$ | $0.066 \pm 0.000$ | $0.917 \pm 0.004$ | $0.055 \pm 0.003$ |

*Table 5.* Additional disentanglement metrics calculated with MINE mutual information estimation for the parametric model dataset. Bold indicates closest to true posterior within group.

| Assumption | Model | $I(y;z)$ | $I(y;w)$ | $I(w;z)$ | $\text{MIG}(w;z)$ | $\text{MIC}(w;z)$ | $I(w;z\mid y)$ |
|---|---|---|---|---|---|---|---|
| | CSVAE - No Adv. | $\mathbf{0.096 \pm 0.037}$ | $0.528 \pm 0.026$ | $\mathbf{0.096 \pm 0.034}$ | $\mathbf{0.048 \pm 0.006}$ | $\mathbf{0.848 \pm 0.057}$ | $0.001 \pm 0.001$ |
| | CSVAE | $0.033 \pm 0.055$ | $\mathbf{0.526 \pm 0.045}$ | $0.031 \pm 0.045$ | $0.055 \pm 0.010$ | $0.945 \pm 0.091$ | $0.009 \pm 0.005$ |
| Fully Separable | HCSVAE - No Adv. | $0.000 \pm 0.000$ | $0.543 \pm 0.018$ | $0.000 \pm 0.000$ | $0.061 \pm 0.002$ | $1.000 \pm 0.000$ | $0.000 \pm 0.000$ |
| | HCSVAE | $0.000 \pm 0.000$ | $0.543 \pm 0.022$ | $0.000 \pm 0.000$ | $0.061 \pm 0.002$ | $1.000 \pm 0.000$ | $0.001 \pm 0.000$ |
| | DisCoVR (CSVAE prior) | $0.020 \pm 0.004$ | $0.543 \pm 0.018$ | $0.033 \pm 0.005$ | $0.058 \pm 0.002$ | $0.964 \pm 0.008$ | $\mathbf{0.030 \pm 0.004}$ |
| | DIVA | $0.027 \pm 0.053$ | $0.113 \pm 0.056$ | $0.001 \pm 0.001$ | $0.010 \pm 0.012$ | $0.798 \pm 0.398$ | $0.001 \pm 0.000$ |
| Flexible | CCVAE | $0.026 \pm 0.052$ | $0.115 \pm 0.058$ | $0.001 \pm 0.001$ | $0.010 \pm 0.012$ | $0.799 \pm 0.399$ | $0.001 \pm 0.000$ |
| | DisCoVR (ours) | $\mathbf{0.037 \pm 0.006}$ | $\mathbf{0.176 \pm 0.008}$ | $\mathbf{0.109 \pm 0.025}$ | $\mathbf{0.016 \pm 0.001}$ | $\mathbf{0.825 \pm 0.026}$ | $\mathbf{0.073 \pm 0.018}$ |
| Fully Separable | Posterior (no truncation) | $0.084 \pm 0.003$ | $0.083 \pm 0.003$ | $0.137 \pm 0.004$ | $0.000 \pm 0.000$ | $0.497 \pm 0.009$ | $0.088 \pm 0.003$ |
| | True Posterior | $0.085 \pm 0.005$ | $0.493 \pm 0.011$ | $0.139 \pm 0.005$ | $0.046 \pm 0.002$ | $0.853 \pm 0.009$ | $0.057 \pm 0.004$ |

*Table 6.* Rank (1 = closest to True Posterior) of each method with respect to the true posterior for metrics calculated with kNN mutual information estimation with $k = 20$. Colors indicate rank within each block: red = worse (farther), green = better (closer).

| Assumption | Model | $I(y;z)$ | $I(y;w)$ | $I(w;z)$ | $\text{MIG}(w;z)$ | $\text{MIC}(w;z)$ | $I(w;z\mid y)$ |
|---|---|---|---|---|---|---|---|
| | CSVAE - No Adv. | 1 | 4 | 1 | 2 | 1 | 3 |
| | CSVAE | 2 | 5 | 3 | 1 | 2 | 2 |
| Fully Separable | HCSVAE - No Adv. | 3 | 1 | 5 | 3 | 3 | 3 |
| | HCSVAE | 3 | 1 | 4 | 3 | 3 | 3 |
| | DisCoVR (CSVAE prior) | 3 | 1 | 2 | 3 | 3 | 1 |
| | DIVA | 2 | 2 | 2 | 2 | 2 | 2 |
| Flexible | CCVAE | 1 | 3 | 2 | 2 | 2 | 2 |
| | DisCoVR (our prior) | 3 | 1 | 1 | 1 | 1 | 1 |

*Table 7.* Rank (1 = closest to True Posterior) of each method with respect to the True Posterior for metrics calculated with MINE mutual information estimation. Colors indicate rank within each block: red = worse (farther), green = better (closer).

| Assumption | Model | $I(y;z)$ | $I(y;w)$ | $I(w;z)$ | $MIG(w;z)$ | $MIC(w;z)$ | $I(w;z\mid y)$ |
|---|---|---|---|---|---|---|---|
| Fully Separable | CSVAE - No Adv. | 1 | 2 | 1 | 1 | 1 | 3 |
| | CSVAE | 2 | 1 | 3 | 2 | 2 | 2 |
| | HCSVAE - No Adv. | 4 | 3 | 4 | 4 | 4 | 5 |
| | HCSVAE | 4 | 3 | 4 | 4 | 4 | 3 |
| | DisCoVR (CSVAE prior) | 3 | 3 | 2 | 3 | 3 | 1 |
| Flexible | DIVA | 2 | 3 | 2 | 2 | 3 | 2 |
| | CCVAE | 3 | 2 | 2 | 2 | 2 | 2 |
| | DisCoVR (ours) | 1 | 1 | 1 | 1 | 1 | 1 |

## E.2. Noisy Swiss Roll

When the observed labels are noisy, DisCoVR outperforms other methods, obtaining embeddings close to the ground truth.

*Table 8.* Additional disentanglement metrics calculated with kNN mutual information estimation for the Noisy Swiss Roll ($p = 0.3$) dataset with $k = 20$. Bold indicates closest to ground truth within group.

| Assumption | Model | $I(y;z)$ | $I(y;w)$ | $I(w;z)$ | $MIG(w;z)$ | $MIC(w;z)$ | $I(w;z\mid y)$ |
|---|---|---|---|---|---|---|---|
| Fully Separable | CSVAE - No Adv. | $0.041 \pm 0.007$ | $0.525 \pm 0.221$ | $0.362 \pm 0.180$ | $0.057 \pm 0.026$ | $0.888 \pm 0.098$ | $0.266 \pm 0.152$ |
| | CSVAE | $0.018 \pm 0.026$ | $\mathbf{0.429 \pm 0.254}$ | $0.240 \pm 0.181$ | $\mathbf{0.048 \pm 0.032}$ | $0.912 \pm 0.129$ | $0.186 \pm 0.146$ |
| | HCSVAE - No Adv. | $0.029 \pm 0.007$ | $0.642 \pm 0.000$ | $0.065 \pm 0.013$ | $0.072 \pm 0.001$ | $0.957 \pm 0.010$ | $0.009 \pm 0.019$ |
| | HCSVAE | $\mathbf{0.001 \pm 0.002}$ | $0.641 \pm 0.001$ | $\mathbf{0.005 \pm 0.004}$ | $0.075 \pm 0.000$ | $\mathbf{0.999 \pm 0.003}$ | $\mathbf{0.000 \pm 0.000}$ |
| Flexible | DIVA | $0.034 \pm 0.013$ | $0.036 \pm 0.011$ | $2.633 \pm 0.360$ | $0.000 \pm 0.003$ | $0.515 \pm 0.159$ | $2.185 \pm 0.332$ |
| | CCVAE | $0.040 \pm 0.015$ | $0.030 \pm 0.007$ | $2.952 \pm 0.124$ | $-0.001 \pm 0.003$ | $0.447 \pm 0.153$ | $2.462 \pm 0.118$ |
| | DisCoVR (ours) | $\mathbf{0.000 \pm 0.000}$ | $\mathbf{0.049 \pm 0.002}$ | $\mathbf{0.029 \pm 0.011}$ | $\mathbf{0.006 \pm 0.000}$ | $\mathbf{1.000 \pm 0.000}$ | $\mathbf{0.014 \pm 0.008}$ |
| Noisy | Ground Truth | $0.000 \pm 0.000$ | $0.055 \pm 0.002$ | $0.000 \pm 0.000$ | $0.007 \pm 0.000$ | $1.000 \pm 0.000$ | $0.000 \pm 0.000$ |

*Table 9.* Additional disentanglement metrics calculated with MINE mutual information estimation for the Noisy Swiss Roll ($p = 0.3$) dataset. Bold indicates closest to ground truth within group.

| Assumption | Model | $I(y;z)$ | $I(y;w)$ | $I(w;z)$ | $MIG(w;z)$ | $MIC(w;z)$ | $I(w;z\mid y)$ |
|---|---|---|---|---|---|---|---|
| Fully Separable | CSVAE - No Adv. | $0.046 \pm 0.022$ | $0.422 \pm 0.186$ | $0.050 \pm 0.020$ | $0.044 \pm 0.023$ | $0.834 \pm 0.184$ | $0.029 \pm 0.017$ |
| | CSVAE | $0.023 \pm 0.027$ | $\mathbf{0.373 \pm 0.232}$ | $0.027 \pm 0.020$ | $\mathbf{0.041 \pm 0.028}$ | $0.877 \pm 0.198$ | $0.024 \pm 0.017$ |
| | HCSVAE - No Adv. | $0.023 \pm 0.014$ | $0.585 \pm 0.011$ | $0.026 \pm 0.012$ | $0.066 \pm 0.002$ | $0.963 \pm 0.021$ | $0.006 \pm 0.002$ |
| | HCSVAE | $\mathbf{0.002 \pm 0.000}$ | $0.570 \pm 0.011$ | $\mathbf{0.002 \pm 0.001}$ | $0.067 \pm 0.001$ | $\mathbf{0.997 \pm 0.001}$ | $\mathbf{0.003 \pm 0.001}$ |
| Flexible | DIVA | $0.041 \pm 0.024$ | $0.043 \pm 0.026$ | $0.313 \pm 0.084$ | $\mathbf{0.000 \pm 0.006}$ | $0.507 \pm 0.296$ | $0.345 \pm 0.065$ |
| | CCVAE | $0.056 \pm 0.020$ | $\mathbf{0.036 \pm 0.020}$ | $0.507 \pm 0.114$ | $-0.002 \pm 0.004$ | $0.390 \pm 0.226$ | $0.494 \pm 0.099$ |
| | DisCoVR (ours) | $\mathbf{0.001 \pm 0.000}$ | $0.069 \pm 0.002$ | $\mathbf{0.004 \pm 0.002}$ | $0.008 \pm 0.000$ | $\mathbf{0.983 \pm 0.004}$ | $\mathbf{0.006 \pm 0.002}$ |
| Noisy | Ground Truth | $0.000 \pm 0.000$ | $0.024 \pm 0.018$ | $0.000 \pm 0.001$ | $0.003 \pm 0.002$ | $0.985 \pm 0.048$ | $0.002 \pm 0.001$ |

*Table 10.* Rank (1 = closest to Ground Truth) of each method with respect to the Ground Truth for metrics calculated with kNN mutual information estimation with $k = 20$ on the Noisy Swiss Roll ($p = 0.3$) dataset. Colors indicate rank within each block: red = worse (farther), green = better (closer).

| Assumption | Method | $I(y;z)$ | $I(y;w)$ | $I(w;z)$ | $MIG(w;z)$ | $MIC(w;z)$ | $I(w;z\mid y)$ |
|---|---|---|---|---|---|---|---|
| Fully Separable | CSVAE - No Adv. | 4 | 2 | 4 | 2 | 4 | 4 |
| | CSVAE | 2 | 1 | 3 | 1 | 3 | 3 |
| | HCSVAE - No Adv. | 3 | 4 | 2 | 3 | 2 | 2 |
| | HCSVAE | 1 | 3 | 1 | 4 | 1 | 1 |
| Flexible | DIVA | 2 | 2 | 2 | 2 | 2 | 2 |
| | CCVAE | 3 | 3 | 3 | 3 | 3 | 3 |
| | DisCoVR (ours) | 1 | 1 | 1 | 1 | 1 | 1 |

*Table 11.* Rank (1 = closest to Ground Truth) of each method with respect to the Ground Truth for metrics calculated with MINE mutual information estimation on the Noisy Swiss Roll ($p = 0.3$) dataset. Colors indicate rank within each block: red = worse (farther), green = better (closer).

| Assumption | Model | $I(y; z)$ | $I(y; w)$ | $I(w; z)$ | $\mathrm{MIG}(w; z)$ | $\mathrm{MIC}(w; z)$ | $I(w; z \mid y)$ |
|---|---|---|---|---|---|---|---|
| Fully Separable | CSVAE - No Adv. | 4 | 2 | 4 | 2 | 4 | 4 |
|  | CSVAE | 2 | 1 | 3 | 1 | 3 | 3 |
|  | HCSVAE - No Adv. | 2 | 4 | 2 | 3 | 2 | 2 |
|  | HCSVAE | 1 | 3 | 1 | 4 | 1 | 1 |
| Flexible | DIVA | 2 | 2 | 2 | 1 | 2 | 2 |
|  | CCVAE | 3 | 1 | 3 | 2 | 3 | 3 |
|  | DisCoVR (ours) | 1 | 3 | 1 | 2 | 1 | 1 |

# F. Ablations on model components

Here we evaluate two variations of our model: (1) applying the classifier directly to $z$, and (2) replacing the conditional prior on $w$ with a standard Gaussian.

When training the classifier directly on $z$ we were able to achieve results qualitatively similar to those obtained using the reconstruction $\hat{x}$, but doing so requires substantially more parameter tuning.

An unconditional standard Gaussian prior for $w$ causes $w$ to collapse into a representation redundant with $z$, removing meaningful separation.

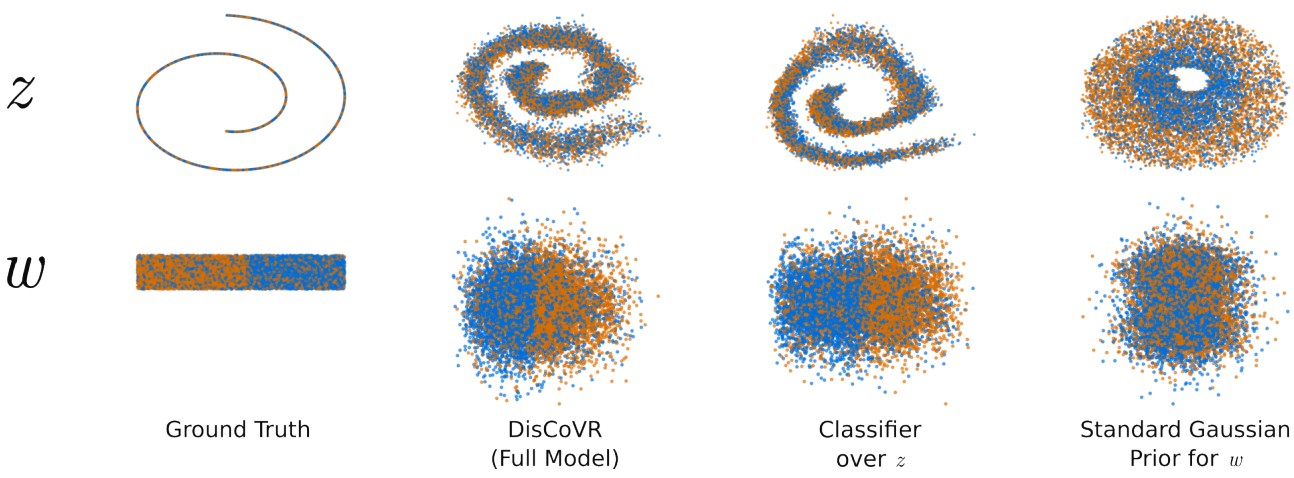

*Figure 7.* Ablation study on the Noisy Swiss Roll ($p = 0.3$) dataset.

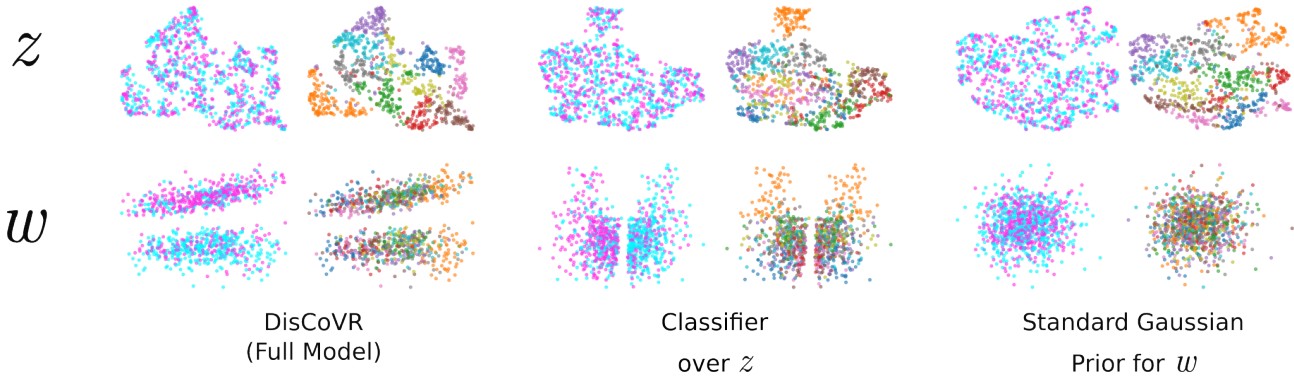

*Figure 8.* Ablation study on the Noisy Colored MNIST ($p = 0.3$) dataset. For each setting: left column denotes coloring by noisy labels, right column denotes coloring by digit (shape, not included in the label).

We consider an additional ablation of the adversarial component, by varying the weight of the adversarial loss on the Noisy Swiss Roll dataset. The results for kNN MI estimation and MINE MI estimation are presented in Tables 12 and 13 respectively.

*Table 12.* Information metrics estimated using kNN mutual information across different adversarial weights for the Noisy Swiss Roll dataset. Arrows indicate the desired direction. Bold denotes best performance.

| | $I(y; z) \downarrow$ | $I(y; w) \uparrow$ | $I(w; z) \downarrow$ | $\text{MIG}(w; z) \uparrow$ | $\text{MIC}(w; z) \uparrow$ | $I(w; z \mid y) \downarrow$ |
|---|---|---|---|---|---|---|
| Adv. = 0 | 0.049 | 0.000 | 0.887 | $-0.006$ | 0.000 | 0.812 |
| Adv. = 2 | 0.051 | 0.000 | 0.489 | $-0.006$ | 0.000 | 0.427 |
| Adv. = 4 | 0.052 | 0.000 | 0.480 | $-0.006$ | 0.000 | 0.419 |
| Adv. = 6 | 0.007 | 0.045 | 0.179 | 0.005 | 0.873 | 0.112 |
| Adv. = 8[†] | **0.000** | **0.050** | **0.032** | **0.006** | **1.000** | **0.017** |

[†] The value used in the paper.

*Table 13.* Information metrics estimated using MINE mutual information across different adversarial weights for the Noisy Swiss Roll dataset. Arrows indicate the desired direction. Bold denotes best performance.

| | $I(y; z) \downarrow$ | $I(y; w) \uparrow$ | $I(w; z) \downarrow$ | $\text{MIG}(w; z) \uparrow$ | $\text{MIC}(w; z) \uparrow$ | $I(w; z \mid y) \downarrow$ |
|---|---|---|---|---|---|---|
| Adv. = 0 | 0.067 | 0.011 | 0.222 | $-0.007$ | 0.136 | 0.249 |
| Adv. = 2 | 0.051 | 0.011 | 0.105 | $-0.005$ | 0.182 | 0.118 |
| Adv. = 4 | 0.051 | 0.011 | 0.101 | $-0.005$ | 0.184 | 0.111 |
| Adv. = 6 | 0.003 | 0.064 | 0.015 | 0.007 | 0.950 | 0.020 |
| Adv. = 8[†] | **0.001** | **0.066** | **0.004** | **0.008** | **0.980** | **0.008** |

[†] The value used in the paper.

## G. Additional experiment on CelebA-Hats

We performed an additional experiment on the CelebA dataset, with the attribute $Wearing\_hat$ denoting the $y$ label. Supplementary Table 14 outlines the results of this experiment. DisCoVR is the only method that exhibits high disentanglement for $z, w$ without compromising reconstruction quality.

*Table 14.* Model performances of a single experiment on CelebA-Hats. Bold denotes best performance.

|  | $I(z;w) \downarrow$ | NLL ($\downarrow$) |
|---|---|---|
| CSVAE - No Adv. | 0.360 | 653.537 |
| CSVAE | 0.213 | **351.082** |
| HCSVAE - No Adv. | 0.135 | 2608.442 |
| HCSVAE | 0.192 | 673.674 |
| DIVA | 0.553 | 356.090 |
| CCVAE | 0.856 | 347.940 |
| DisCoVR (ours) | **0.059** | 353.271 |
| DisCoVR (ours) - Common | - | 437.144 |

## H. Implementation details

### H.1. Considerations and reproducibility

We run all experiments on a single H100 GPU. Reported means and standard deviations for tables are conducted over 10 repetitions of the experiment with different random seeds. All models are trained using the AdamW (Loshchilov & Hutter, 2019) optimizer until validation loss stops decreasing for 50 epochs. Wherever provided, we use mutual information neural estimation (MINE, Belghazi et al. (2018)) and k-Nearest Neighbor (kNN) mutual information estimation (Kraskov et al., 2004) to obtain mutual information estimates. For Naive Bayes classifiers, we use the implementation provided by *scikit-learn* (Pedregosa et al., 2011). To use ideal hyperparameters for each method, we consult the original implementation whenever possible, and conduct a simple grid-search to produce originally described model behavior. Implementations of all methods compared in this study, including DisCoVR, as well as code to reproduce our results, are available at `https://github.com/Computational-Morphogenomics-Group/DISCoVeR`. Models compared in the study admit a weighting term for each term in the loss function, of which most are shared across different approaches. We use the following shorthands for each of the terms:

$$\text{Rec.} \rightarrow \mathbb{E}_{q_{z|x}} \left[ \mathbb{E}_{q_{w|x,y}}[\log p(x \mid z, w)] \right]$$
$$\mathcal{D}_{\text{KL}}(Z) \rightarrow \mathcal{D}_{\text{KL}}(q_{z|x} \| p_z)$$
$$\mathcal{D}_{\text{KL}}(W) \rightarrow \mathcal{D}_{\text{KL}}\left(q_{w|x,y} \| p_{w|y}\right)$$
$$\text{Adv.} \rightarrow -\mathbb{E}_{q_{z|x}}[\log g(y \mid z)]$$
$$\text{Class.} \rightarrow \mathbb{E}_{q_{w|x,y}}[\log q(y \mid w)]$$
$$\text{Rec. - }(Z) \rightarrow \mathbb{E}_{q_{z|x}}[\log p(x \mid z)]$$

Below, we provide additional details for the hyperparameters used in each experiment, and any other external resources used to obtain the corresponding sections' results. In addition, we include details regarding runtime and memory footprint of running experiments with the models included in our study.

*Table 15.* Time spent per epoch during training for each dataset.

|  | P.M. | N.S.R | CMNIST | CelebA | scRNA-seq |
|---|---|---|---|---|---|
| CSVAE - No Adv. | 10.91s | 8.02s | 14s | 54.43s | 4.78s |
| CSVAE | 12.71s | 8.98s | 14.41s | 74.68s | 4.99s |
| HCSVAE - No Adv. | 15.6s | 10.92s | 17s | 44.3s | 6.34s |
| HCSVAE | 16.51s | 11.8s | 16.8s | 68.53s | 5.3s |
| DIVA | 11.1s | 8s | 18.59s | 48.96s | 3.55s |
| CCVAE | 12.1s | 8.44s | 17.9s | 49.65s | 5.25s |
| DisCoVR(ours) | 12.18s | 8.73s | 21.8s | 109.86s | 6.09s |

*Table 16.* Model inference time for a single batch for each dataset.

|  | P.M. | N.S.R | CMNIST | CelebA | scRNA-seq |
|---|---|---|---|---|---|
| CSVAE - No Adv. | 72ms | 28ms | 57ms | 329ms | 100ms |
| CSVAE | 40ms | 29ms | 59ms | 187ms | 96ms |
| HCSVAE - No Adv. | 36ms | 30ms | 55ms | 214ms | 95ms |
| HCSVAE | 37ms | 39ms | 52ms | 191ms | 119ms |
| DIVA | 25ms | 26ms | 60ms | 154ms | 42ms |
| CCVAE | 27ms | 28ms | 42ms | 163ms | 93ms |
| DisCoVR(ours) | 32ms | 27ms | 63ms | 205ms | 123ms |

*Table 17.* Memory footprint of running an experiment for each dataset.

|  | P.M. | N.S.R | CMNIST | CelebA | scRNA-seq |
|---|---|---|---|---|---|
| CSVAE - No Adv. | 53 MiB | 255 MiB | 1988 MiB | 4868 MiB | 298 MiB |
| CSVAE | 253 MiB | 255 MiB | 2378 MiB | 4812 MiB | 300 MiB |
| HCSVAE - No Adv. | 254 MiB | 255 MiB | 2558 MiB | 4588 MiB | 292 MiB |
| HCSVAE | 253 MiB | 256 MiB | 2998 MiB | 4466 MiB | 294 MiB |
| DIVA | 253 MiB | 255 MiB | 2634 MiB | 4996 MiB | 300 MiB |
| CCVAE | 253 MiB | 255 MiB | 3066 MiB | 4998 MiB | 300 MiB |
| DisCoVR (ours) | 254 MiB | 257 MiB | 3612 MiB | 7078 MiB | 308 MiB |

### H.1.1. PARAMETRIC MODEL

For the parametric model, we consider $z, w \in \mathbb{R}$ and use multi-layer perceptrons (MLPs) with $n_{hidden} = 2, d_{hidden} = 8$ to parameterize approximate posteriors, the generative model and classifiers. For all models, we use learning rate $\gamma = 0.001$. A more detailed table of model-specific loss weights is provided in Supplementary Table 18.

*Table 18.* Loss weights for the parametric model experiment.

|  | Rec. | $\mathcal{D}_{\mathrm{KL}}(Z)$ | $\mathcal{D}_{\mathrm{KL}}(W)$ | Adv. | Class. | Rec. - $(Z)$ |
|---|---|---|---|---|---|---|
| CSVAE - No Adv. | 1 | 1 | 1 | — | — | — |
| CSVAE | 2.5 | 1 | 0.5 | 20 | — | — |
| HCSVAE - No Adv. | 1 | 1 | 0.5 | — | — | — |
| HCSVAE | 2.5 | 1 | 0.5 | 20 | — | — |
| DIVA | 1 | 1 | 1 | — | 1 | — |
| CCVAE | 1 | 1 | 1 | — | 1 | — |
| DisCoVR (ours) | 0.75 | 0.9 | 0.2 | 0.8 | — | 0.25 |

*Table 19.* K-Means NMI for embeddings across stimulation ($y$) and cell type (common structure).

|  | $w$ - Stimulation ($\uparrow$) | $z$ - Cell Type ($\uparrow$) | $z$ - Stimulation ($\downarrow$) |
|---|---|---|---|
| CSVAE - No Adv. | **0.949 ± 0.003** | 0.702 ± 0.015 | 0.187 ± 0.0 |
| CSVAE | 0.939 ± 0.002 | 0.406 ± 0.001 | **0.002 ± 0.0** |
| HCSVAE - No Adv. | 0.933 ± 0.006 | 0.628 ± 0.016 | 0.091 ± 0.002 |
| HCSVAE | 0.931 ± 0.005 | 0.433 ± 0.001 | 0.003 ± 0.0 |
| DIVA | 0.801 ± 0.0 | 0.628 ± 0.011 | 0.056 ± 0.0 |
| CCVAE | 0.604 ± 0.0 | 0.683 ± 0.016 | 0.103 ± 0.0 |
| DisCoVR (ours) | 0.906 ± 0.003 | **0.716 ± 0.031** | **0.002 ± 0.001** |

### H.1.2. NOISY SWISS ROLL

For this experiment, we consider $z, w \in \mathbb{R}^2$ and use MLPs with $n_{hidden} = 2, d_{hidden} = 128$ to parameterize approximate posteriors, the generative model and classifiers. For all models, we use learning rate $\gamma = 0.001$. A more detailed table of model-specific hyperparameters is provided in Supplementary Table 20.

*Table 20.* Loss weights for the noisy Swiss roll experiment.

|  | Rec. | $\mathcal{D}_{\mathrm{KL}}(Z)$ | $\mathcal{D}_{\mathrm{KL}}(W)$ | Adv. | Class. | Rec. - ($Z$) |
|---|---|---|---|---|---|---|
| CSVAE - No Adv. | 20 | 0.2 | 1 | — | — | — |
| CSVAE | 20 | 0.2 | 1 | 50 | — | — |
| HCSVAE - No Adv. | 20 | 0.2 | 1 | — | — | — |
| HCSVAE | 20 | 0.5 | 1 | 50 | — | — |
| DIVA | 20 | 0.2 | 0.2 | — | 1 | — |
| CCVAE | 20 | 0.2 | 0.2 | — | 1 | — |
| DisCoVR (ours) | 0.9 | 0.2 | 0.2 | 8 | — | 0.1 |

### H.1.3. NOISY COLORED MNIST

For this experiment, we consider $z \in \mathbb{R}^{20}$, $w \in \mathbb{R}^2$ and use convolutional neural networks (CNNs) to parameterize approximate posteriors and the generative model. For this example, DisCoVR can support $z, w$ with different sizes, by parameterizing $p(w \mid y)$ through neural networks. For all models, we use learning rate $\gamma = 0.0001$. We detail the architectures and model-specific hyperparameters in Supplementary Tables 21 - 24. All other neural networks are formulated as MLPs with $n_{hidden} = 2, d_{hidden} = 4096$.

*Table 21.* Image encoder architecture for noisy colored MNIST. Parameters for Conv2d are input / output channels. Parameters for MaxPool2D are kernel size and stride. Parameter for the linear layer is the output size. For variances, outputs are passed through an additional Softplus layer to ensure non-negativity.

| Block | Details |
|---|---|
| 1 | Conv2d(3,32) + BatchNorm2D + ReLU |
| 2 | Conv2d(32,32) + BatchNorm2D + ReLU + MaxPool2D(2,2) |
| 3 | Conv2d(32,64) + BatchNorm2D + ReLU + MaxPool2D(2,2) |
| 4 | Conv2d(64,128) + BatchNorm2D + ReLU + MaxPool2D(2,2) |
| 5 | Linear(4096) + BatchNorm1D + ReLU |
| 6 | Linear(4096) + BatchNorm1D + ReLU |
| 7 | Linear($d_{latent}$) |

*Table 22.* Image decoder architecture for noisy colored MNIST. Parameters for Conv2d are input / output channels. Parameters for MaxPool2D are kernel size and stride. Parameter for the linear layer is the output size.

| Block | Details |
|-------|---------|
| 1 | Linear(4096) + BatchNorm1D + ReLU |
| 2 | Linear(4096) + BatchNorm1D + ReLU |
| 3 | Linear(1152) + Unflatten |
| 4 | Upsample(2) + Conv2d(128, 64) + BatchNorm2D + ReLU |
| 5 | Upsample(2) + Conv2d(64, 32) + BatchNorm2D + ReLU |
| 6 | Upsample(2) + Conv2d(32, 32) + BatchNorm2D + ReLU |
| 7 | Conv2d(32, 3) + Sigmoid |

*Table 23.* Latent classifier architecture for noisy colored MNIST. Outputs parameterize logits of class probabilities.

| Block | Details |
|-------|---------|
| 1 | Linear(4096) + BatchNorm1D + ReLU |
| 2 | Linear(4096) + BatchNorm1D + ReLU |
| 3 | Linear(2) |

*Table 24.* Loss weights for the noisy colored MNIST experiment.

|  | Rec. | $\mathcal{D}_{\mathrm{KL}}(Z)$ | $\mathcal{D}_{\mathrm{KL}}(W)$ | Adv. | Class. | Rec. - $(Z)$ |
|---|---|---|---|---|---|---|
| CSVAE - No Adv. | 1 | 0.0001 | 1 | — | — | — |
| CSVAE | 1 | 0.0001 | 1 | 1 | — | — |
| HCSVAE - No Adv. | 1000 | 0.0001 | 1 | — | — | — |
| HCSVAE | 10000 | 0.0001 | 1 | 1 | — | — |
| DIVA | 1 | 0.0001 | 0.0001 | — | 1 | — |
| CCVAE | 1 | 0.0001 | 0.0001 | — | 1 | — |
| DisCoVR (ours) | 0.5 | 0.0001 | 0.0001 | 0.1 | — | 0.5 |

### H.1.4. CELEBA-GLASSES

Motivated by the previous application of Klys et al. (2018), our choices follow those outlined in Larsen et al. (2016). We provide a detailed table of model-specific hyperparameters in Supplementary Table 25:

*Table 25.* Loss weights for the CelebA-Glasses experiment.

|  | Rec. | $\mathcal{D}_{\mathrm{KL}}(Z)$ | $\mathcal{D}_{\mathrm{KL}}(W)$ | Adv. | Class. | Rec. - $(Z)$ |
|---|---|---|---|---|---|---|
| CSVAE - No Adv. | 1 | 0.0001 | 1 | — | — | — |
| CSVAE | 1000 | 0.0001 | 1 | 1 | — | — |
| HCSVAE - No Adv. | 1000 | 0.0001 | 1 | — | — | — |
| HCSVAE | 10000 | 0.0001 | 1 | 1 | — | — |
| DIVA | 100000 | 0.0001 | 0.0001 | — | 1 | — |
| CCVAE | 100000 | 0.0001 | 0.0001 | — | 1 | — |
| DisCoVR (ours) | 1000000 | 0.0001 | 0.0001 | 2000 | — | 100000 |

## H.1.5. scRNA-Seq

Following on the previous applications by Lopez et al. (2018), we use $z \in \mathbb{R}^{10}$, $w \in \mathbb{R}^2$. Similar to the Noisy Colored MNIST example, we use DisCoVR with matched sizes by parameterizing $p(w \mid y)$ through a neural network. We use MLPs with $n_{hidden} = 1, d_{hidden} = 128$ to parameterize approximate posteriors, the generative model and classifiers. We calculate K-Means NMI through *scikit-learn* (Pedregosa et al., 2011) by calling the `normalized_mutual_info_score` function with the original labels and the clusterings obtained by running KMeans on (1) the entire latent embedding and (2) single dimensions of the embedding and report the highest score. A more detailed table of model-specific hyperparameters is provided in Supplementary Table 26:

*Table 26.* Loss weights for the scRNA-seq experiment.

|  | Rec. | $\mathcal{D}_{\mathrm{KL}}(Z)$ | $\mathcal{D}_{\mathrm{KL}}(W)$ | Adv. | Class. | Rec. - $(Z)$ |
|---|---|---|---|---|---|---|
| CSVAE - No Adv. | 1 | 0.0001 | 1 | — | — | — |
| CSVAE | 1 | 0.0001 | 1 | 100 | — | — |
| HCSVAE - No Adv. | 1 | 0.0001 | 1 | — | — | — |
| HCSVAE | 1 | 0.0001 | 1 | 100 | — | — |
| DIVA | 1 | 0.0001 | 0.0001 | — | 1 | — |
| CCVAE | 1 | 0.0001 | 0.0001 | — | 1 | — |
| DisCoVR (ours) | 0.9 | 0.0001 | 0.0001 | 100 | — | 0.1 |

## H.2. Summary of the scVI generative model for 5.2.3

Given batch key $b$ and $G$ genes, the generative model of scVI for a single cell $x_i \in \mathbb{N}^G$ is formulated as:

$$z_i \sim \mathcal{N}(0, 1)$$
$$\rho_i = f_\theta(z_i, b_i)$$
$$\pi_{ig} = h_\phi^g(z_i, b_i)$$
$$x_{ig} \sim \mathrm{ZINB}(l_i\rho_i, \theta_g, \pi_{ig})$$

Here, $g$ indexes genes, $l_i = \sum_g x_{ig}$ denotes the total number of counts for a single cell, $z_i$ denotes the latent representation of the cell, and $\rho_i$ denotes the normalized expression of the cell. $f_\theta$ is formulated as a neural network with a final softmax layer. $h_\phi$ is a neural network used to parameterize zero-inflation probabilities for the generative zero-inflated negative binomial (ZINB) distribution. As such, for a single batch, the formulation of scVI is equivalent to the VAE with a ZINB likelihood. While all other models can be extended easily, DisCoVR requires reconstructions as a proxy for the adversarial loss. For this formulation, we directly treat the normalized expressions $\rho_i$ as the adversarial reconstructions $\hat{x}$.

