# OpenReview forum: "Variational Learning of Disentangled Representations"
_ICML.cc/2026/Conference — ICML 2026 regular_

### Official Review · Reviewer_qTxA · 2026-03-04

**Soundness:** 3
**Presentation:** 3
**Significance:** 2
**Originality:** 3
**Overall Recommendation:** 4
**Confidence:** 2

**Summary:**

This paper introduces DisCoVR, a framework designed to learn disentangled representations in datasets that have multiple conditions. It specifically addresses the common problem of leakage between shared features and condition-specific features found in other models. The authors achieve this by using two separate latent variables: a condition-aware variable $w$ and a shared latent variable $z$. To ensure these stay separate and informative, they implement two different reconstruction paths and an adversarial classifier, which works to remove any condition-related information out of $z$.

**Compliance With Llm Reviewing Policy:**

Affirmed.

**Final Justification:**

DisCoVR presents a well-motivated probabilistic framework for disentangled representation learning. The rebuttal addressed concerns about the adversarial classifier's necessity and stability, supported by empirical evidence showing the classifier is critical for achieving disentanglement.

**Key Questions For Authors:**

- How sensitive is the model with respect to the hyperparameters, in particular $\alpha_1$  and $\alpha_2$ ? Since balancing regular VAEs can already be delicate, I'm wondering if this setup becomes much more complex to deal with in terms of the engineering.
- Could you provide a more detailed breakdown of the actual gains from the classifier? It would be helpful to see how it performs specifically against a version of the model that only uses the dual reconstruction paths and the prior setup.
- What happens to the anchored prior on $w$ if the conditions aren't clearly separated into discrete classes, but are instead based on a continuous score?

**Limitations:**

yes

**Strengths And Weaknesses:**

Strengths
- Probabilistic Approach: The motivation for the method is well-reasoned in a probabilistic setup. The authors clearly describe the graphical model they want to achieve and use that logic to build the entire mathematical setup.
- Latent Coupling: I like the way the condition-specific latent ($w$) is linked to the mean of the shared latent ($z$), but not the reverse direction. By basing the prior of $w$ on the average of $z$ for a specific condition, the model creates a setup that keeps the information flowing from $z$ to $w$ (via $\mu_k$) without letting the categories get mixed up.
- Diverse Results: The model is tested across a variety of challenges, ranging from simple synthetic datasets to more complex tasks like CelebA, color-MNIST, and single-cell RNA sequencing. These results provide good evidence that DisCoVR is successful at untangling these different types of information.

Weaknesses
- More Complexity and Stability: While the reason for adding a separate adversarial classifier is easy to understand, it adds a lot of extra weight to a system that already has an objective with several terms. Training these types of models where different components are competing against each other usually makes the whole process much more fragile. I am curious to know if the actual improvements in practical setups are worth the extra training time and the difficulty of getting the settings exactly right.
- Indirect Classifier: I am not yet entirely convinced by the argument for having the classifier look at the reconstructed image rather than the internal code itself. While they authors state that this gives more stability, it also relies on a very strong decoder. There seems to be a risk that the process of turning that code back into an image might "wash away" some of the leaked information. If that happens, the classifier might be misled into thinking the code is clean, when actually the specific information is still hidden in the latent space but just didn't make it through the final decoding step.

---

> ### Author Rebuttal · Authors · 2026-03-29
>
> Dear reviewer,
>
> Thank you for your thoughtful review.
>
> **Weaknesses:**
>
> Removing the classifier entirely would recover an ELBO objective analogous to the one used in DIVA, though not with the same distributional specification, in particular because of our prior on $w$, and without additional reconstruction from $z$. Our specification (before adding the classifier) provides improvement over DIVA, but in the absence of an explicit mechanism preventing $y$-relevant information from leaking into $z$, full disentanglement is hard to achieve. Please see empirical results in our answer to question 2.
>
> However, as you mention, adding a classifier $g(y \mid z)$ would also lead to similar additional complexity. Exactly for this reason, we apply the classifier to the reconstruction $\hat{x}$ rather than directly to $z$. This means the classifier itself can remain very simple; in most of our experiments it is just multinomial logistic regression on $\hat{x}$.
> Appendix F shows that this simplification does not qualitatively change the results.
>
> Regarding the concern that the model may rely on the decoder, rather than the encoder, to remove $y$-relevant information, the objective provides a strong disincentive for this: If $y$-relevant information is still present in $z$ and helps reconstruct $x$, then suppressing it at the decoder stage directly conflicts with the reconstruction term.
> Empirically, this is consistent with what we observe in the synthetic experiments, where the mutual information $I(y;z)$ can be computed explicitly and therefore measures label information present in the encoder representation itself. Appendix E.1 and E.2 show that $I(y;z)$ is nearly zero, so the information is already removed in $z$, before invoking the decoder.
>
> **Questions:**
>
> 1. Like other VAE-based methods, DisCoVR does require tuning of the loss weights, and it is a bit hard to quantify how much. We did not experience that it requires more tuning than other VAE based methods: in the experiments, we selected hyperparameters by a simple grid search, in the same way as for the baselines, and trained all models with the same optimizer and early-stopping criterion.
>
> 2. As you asked, we performed an experiment showing what happens when the classifier is removed or weakened. We include here results for the swiss-role experiment, where we can compute information metrics on the embedding space. As you can see, the classifier is needed for the method's effectiveness, which is expected, as it follows directly from the probabilistic model, and enforces the structural assumed constraint $z \perp y$ .
>
> Using KNN:
> || I(y;z) $\downarrow$| I(y;w) $\uparrow$ | I(w;z) $\downarrow$| MIG(w;z) $\uparrow$ | MIC(w;z) $\uparrow$ | I(w;z) \| y $\downarrow$
> |--|--|--|--|--|--|--|
> |Adv. = 0| 0.049 | 0.000 |  0.887 | -0.006 | 0.000 | 0.812 |
> |Adv. = 2| 0.051 |  0.000 |  0.489 | -0.006 |  0.000 |  0.427 |
> |Adv. = 4| 0.052 | 0.000 | 0.480 | -0.006 | 0.000 | 0.419 |
> |Adv. = 6| 0.007 |  0.045 | 0.179 | 0.005 | 0.873 | 0.112 |
> |Adv. = 8$^†$| 0.000 | 0.050 | 0.032 | 0.006 | 1.000 | 0.017 |
>
> Using MINE:
> || I(y;z) $\downarrow$| I(y;w) $\uparrow$ | I(w;z) $\downarrow$| MIG(w;z) $\uparrow$ | MIC(w;z) $\uparrow$ | I(w;z) \| y $\downarrow$
> |--|--|--|--|--|--|--|
> |Adv. = 0| 0.067 | 0.011 | 0.222 | -0.007 | 0.136 | 0.249 |
> |Adv. = 2| 0.051 | 0.011 | 0.105 | -0.005 | 0.182 | 0.118 |
> |Adv. = 4| 0.051 | 0.011 |  0.101 | -0.005 | 0.184 | 0.111 |
> |Adv. = 6| 0.003 | 0.064 | 0.015 | 0.007 | 0.950 | 0.020 |
> |Adv. = 8$^†$| 0.001 | 0.066 | 0.004 | 0.008 | 0.980 | 0.008 |
>
> $^†$: The value used in the paper.
>
> 3. At a technical level, extending the anchored prior to continuous conditions would mainly require replacing exact classes by a discretization of $y$. For example, one could partition the range of $y$ into bins, possibly quite fine-grained, using quantiles so that each bin contains enough samples to estimate the corresponding class-level quantities reliably (for example, as you would do when estimating expected calibration error).
>
>    That said, our paper explicitly considers discrete $y$ (please see line 46, left column). This is not because the prior itself fundamentally requires discreteness, but because the notion of disentanglement we study is most natural when $y$ indexes distinct conditions or groups. If $y$ is continuous, the idea of “disentangling from $y$” becomes less clear. In that setting, one would typically not seek to remove information about its value altogether, but aim to model how variation in $y$ is reflected in the data. This is also the reason why even OOD settings study discrete environments (covered in Appendix A.2).

---

> > ### Author Rebuttal · Reviewer_qTxA · 2026-04-03
> >
> > I want to thank the authors for addressing all my questions and for their thorough rebuttal.

---

### Official Review · Reviewer_EbAN · 2026-03-07

**Soundness:** 3
**Presentation:** 4
**Significance:** 2
**Originality:** 3
**Overall Recommendation:** 5
**Confidence:** 5

**Summary:**

The learning of disentangled representations is a cornerstone of AI research since such representations should in principle allow for the generalization to new domains or tasks. There has been a plethora of proposals mostly based on variational autoencoders and approaches from nonlinear independent component analysis to solve the problem with various levels of success. The paper proposes to fix some of the shortcomings of established approaches by introducing adversarial terms and structured priors. The usefulness of the approach is demonstrated across various case studies.

**Compliance With Llm Reviewing Policy:**

Affirmed.

**Final Justification:**

Good paper. Clear accept. I have no suggestions to really improve the paper. I do not think additional experiments are needed or would be helpful. Writing is really good. The reason I give no higher score than accept is that it is not on its own setting a new standard or revolutionary (nor does it need to be for an acceptance).

**Key Questions For Authors:**

see above

code is in the supplement right now so I assume it will be shared after a potential acceptance of the paper since it writes in the readme that it will be released as a package?

**Limitations:**

Like all probabilistic approaches they are quite technical, the priors and variational families are restrictive etc but nothing which is particular to the paper. If at all one could say that the latent bottleneck approaches are maybe less modern now then they were a couple of yeas ago.

**Strengths And Weaknesses:**

The learning of disentangled representations is inherently ill posed. That is why additional assumptions are needed. Motivated by results from nonlinear ICA many approaches have focused on multi view or multi environment assumptions. This is similarly done here.

The paper follows a nice and self contained technical derivation in section 2 (nicely done!) and is a text book example for a case study of using priors and probablistic modeling. This is actually very nice and a joy to read as is the comparison to established work in section 4. There could be some more content for multi-view approaches but it is already rare to find a paper which not just lists related work but actually describes the difference on a technical level. This is all very strong and the experimental section is extensive. I still do not understand what the UMAPs in single cell analysis should actually show and demonstrate but that is a problem of the single cell community and unrelated to the evaluation of the work.

I would still be interested to hear how the authors would scale the approach to higher dimensional latents and in particular if they have views on the move of the field away from low dimensional and bottleneck representations to high dimensional representations e.g. see Saining Xie groups's work on high dimensional, patch / token level representations from Dino / SIgLip / MAE like Representation Autoencoders (RAEs) [1]


[1] https://arxiv.org/abs/2510.11690

---

> ### Author Rebuttal · Authors · 2026-03-29
>
> Dear reviewer,
>
> Thank you for your review! We truly enjoyed reading your positive feedback and your question.
>
> Indeed, the work of Saining Xie’s group, and some others, moves from VAE latent spaces to semantic representations, usually pre-trained with diffusion.
> Two key arguments in favor of this are that (i) low dimensional representations are bottlenecks of information, and (ii) that richer representation spaces may organize the data according to more meaningful structure. Both of these are very good arguments, but their relevance depends on the goal for which we are learning the representation.
>
> Whether dimensionality is a bottleneck depends on the task, and on the granularity of detail we aim to preserve for that task. The more important argument is about richness: if the purpose of representation learning is, for example, for causal analysis and identifying stable underlying factors, then interpretability of the latent space is crucial to understand which parts of the representation correspond to which "real world" concepts. But if the goal is, for example, to generate samples that remove condition-specific variation, then direct interpretability of the latent coordinates may be less important than supporting this, which is harder to do with pretrained representations.
>
> In short, we think that the answer to which kind of the representation is preferable depends on the end-goal. We would be very interested to hear your thoughts on this as well!
>
> **Regarding UMAP:** We share your reservations regarding the use of UMAP  in single cell analysis. Here specifically, row A is colored by cell type, and is meant to answer whether a representation preserves biological cell-type structure.
> Since cells that have the same color appear together in $z$, it shows that $z$ indeed preserves the shared cell-type structure.
>
> Row B is colored by condition, so it aims to answer whether a representation preserves stimulation information or removes it.
> In $z$ both colors appear in each cluster, meaning $z$ removes stimulation information. In $w$ there is a clear separation by color, showing that $w$ preserves this information.
>
> **Regarding Code:** Yes, the code will of course be publicly available.

---

> > ### Author Rebuttal · Reviewer_EbAN · 2026-04-03
> >
> > I have read the other reviews and rebuttals. It is a good paper and should be accepted. If there are remaining concerns by other reviewers I would argue that these are based on taste and cosmetics rather than being a fundamental issue.  I do not think that an additional experiment or something else would improve the paper now. The reason I do not give a higher score than accept is that I think it is very nicely done but not necessarily setting a new research direction or contradicting existing results (nor does the paper claim that or have to claim that for an accept).  I have updated my confidence to indicate that from 4 to certain and would recommend acceptance in its current form.
> >
> > On the side note: The open issue with VAE based disentanglement is that the community has been focused on causal representation learning and or interpretability for which these methods seem interesting usecases. Yet both for causal representation learning or interpretability we have not necessarily seen empirical results that indicate that these methods are helpful for downstream tasks (generalization, transfer, sample efficiency) beyond the sake of perceived interpretation. For any policy trained on top of disentangled representations it seems that representation quality matters more than its structure. That is why it might be promising to look more into high dimensional representations in any case.

---

### Official Review · Reviewer_7vqP · 2026-03-12

**Soundness:** 2
**Presentation:** 3
**Significance:** 2
**Originality:** 3
**Overall Recommendation:** 3
**Confidence:** 3

**Summary:**

This paper proposes **DisCoVR**, a supervised variational framework for disentangling multi-condition data into a condition-invariant latent variable `z` and a condition-aware latent variable `w`. The training objective combines: (i) a reconstruction path from `z` alone, (ii) a full reconstruction path from `(z, w)`, (iii) KL regularization terms, and (iv) an adversarial predictor intended to remove label information from `z`. A distinctive modeling choice is a class-conditional prior on `w` whose mean is tied to the class-wise average of inferred `z`. The paper also gives a variational motivation for the factorized posterior approximation, proves a gap expression for using `q(z|x)q(w|x,y)` instead of a full posterior, and presents an idealized saddle-point result under convexity/regularity assumptions. Empirically, the method is evaluated on an analytic parametric model, noisy Swiss roll, noisy colored MNIST, CelebA (glasses and hats), and single-cell RNA-seq, with reported improvements over CSVAE/HCSVAE, DIVA, and CCVAE on several disentanglement-oriented metrics while keeping reconstruction competitive.

**Compliance With Llm Reviewing Policy:**

Affirmed.

**Key Questions For Authors:**

1. **What is the exact scope of Proposition 2.2 relative to the implemented model?**
   The proposition assumes convex/compact variational families and classifier families, but the actual model uses neural-network-parameterized diagonal Gaussians and neural classifiers. Please clarify whether the theorem is intended only as an idealized population-level result. If you can either (a) restate the theorem much more explicitly as such, or (b) provide a result closer to the actual model class, that would improve my soundness assessment.

2. **Is the full training objective meant to be an auxiliary regularized objective or an ELBO for a single probabilistic model?**
   `L_w` is clear, but `L_z` introduces a separate `p(x|z)` term that does not obviously belong to the same graphical model as `p(x|z,w)`. Please clarify whether this is an auxiliary regularizer by design or whether there is a principled derivation from a single unified model. A convincing clarification would materially improve my view of the method’s probabilistic grounding.

3. **Can you provide matched-capacity comparisons, especially for the scRNA-seq experiment?**
   In the appendix, DisCoVR appears to use a larger `w` dimensionality than the baselines in at least one experiment. Please rerun the relevant results with matched latent dimensionalities, or explain why this is not a real capacity mismatch. If the empirical advantage remains under matched capacity, that would strengthen my evaluation.

4. **Can you add direct quantitative tests of label leakage and shared-factor preservation on the real datasets?**
   For example: post-hoc prediction of `y` from `z`, prediction of common structure from `w`, or a downstream transfer/generalization task. This would substantially strengthen the real-data evidence and could improve my significance score.

5. **How are the class means `μ_y` estimated during training and how stable is this under class imbalance / many conditions?**
   Please clarify whether these are computed using full-dataset statistics, mini-batches, or moving averages, and provide sensitivity evidence if possible. This would increase my confidence in the method’s robustness.

**Limitations:**

No. The paper does not adequately discuss several important limitations:
(i) the method is supervised and requires condition labels;
(ii) the structured prior depends on class-wise encoder statistics and may be sensitive to class imbalance, many conditions, or unstable early training;
(iii) the saddle-point theorem does not cover the actual neural parameterization used in experiments;
(iv) the broader claim about improved generalization to unseen conditions is not directly evaluated; and
(v) the proposed method can have noticeably higher compute/memory cost than baselines. I would encourage the authors to discuss these points explicitly.

**Strengths And Weaknesses:**

## Strengths And Weaknesses*

### Strengths

1. **Important problem setting.** Learning representations that separate shared structure from condition-specific variation is a meaningful problem with relevance to domain adaptation, scientific data integration, and transfer settings. The paper is well motivated in that sense.

2. **Interesting method design.** The combination of a `z`-only reconstruction path, a `(z,w)` reconstruction path, adversarial removal of `y` from `z`, and a class-dependent prior on `w` tied to class-wise statistics of `z` is a nontrivial and reasonably original design. Even if each ingredient is not entirely new on its own, the combination is thoughtful and empirically appears effective.

3. **Strong controlled experiments.** The synthetic parametric model with an analytically characterized posterior is a particularly good choice. It gives a setting where the paper can compare learned posteriors to known targets rather than relying only on proxy metrics. The noisy Swiss roll setup is also a useful stress test because it makes the marginal/conditional structure visually interpretable.

4. **Broad empirical scope and decent reproducibility effort.** The paper includes synthetic, image, and biological data, reports means/std over multiple seeds, provides ablations, and includes implementation details plus runtime/memory tables in the appendix. This is better than many submissions in this area.

5. **Readable overall narrative.** The paper is organized sensibly: motivation, model, comparison to prior VAE-based approaches, experiments, and supplementary details. I did not find it hard to follow the high-level story.

### Weaknesses

1. **The main theoretical guarantee does not appear to apply to the implemented model class.** Proposition 2.2 assumes convex/compact variational families and a convex classifier family. But the actual instantiated families are neural-network-parameterized diagonal Gaussians and neural classifiers. These classes are not convex in the sense required by the proposition, and the Gaussian family used in practice is not compact either. As written, the “unique saddle point” result is therefore an idealized functional statement, not a guarantee for the actual training problem. This substantially weakens the paper’s soundness relative to how strongly the theorem is framed.

2. **The objective is only partly “principled probabilistic,” and this should be stated more carefully.** The `L_w` term is a standard ELBO for the factorized posterior under the stated generative model. However, `L_z` introduces an additional `p(x|z)` reconstruction path that is not part of the same graphical model `p(y)p(w|y)p(z)p(x|z,w)`. That auxiliary term may be a very reasonable regularizer, but the resulting training criterion is not obviously a single ELBO of the stated model. In other words, part of the objective is principled variational inference, and part is an additional representation-shaping regularizer. The manuscript currently overstates the probabilistic alignment.

3. **Some theoretical claims are too strong relative to the mechanism actually used.** In particular, the paper argues that the structured prior prevents reverse leakage from `w` into `z`. I do not think this is fully justified as written, because the prior mean `μ_y` is itself a function of class-wise encoder outputs from `z`, so optimization still couples `z` and `y`. The adversarial term seems to be doing essential work here. I would like a more careful account of what is guaranteed versus what is empirically encouraged.

4. **The real-data evidence is weaker than the synthetic evidence.** On noisy colored MNIST and CelebA, the case for disentanglement relies heavily on visual inspection. The main paper emphasizes `I(z;w)` and reconstruction quality, but low `I(z;w)` is not the same thing as showing that `z` is label-invariant and `w` captures the label-specific effect. Direct post-hoc tests such as predicting `y` from `z`, predicting shared factors from `w`, or downstream transfer tasks would make the real-data evidence much more convincing.

5. **The significance claim is broader than what is empirically demonstrated.** The paper motivates the method by potential generalization to new domains/treatments/patients/species, but the experiments do not actually evaluate transfer to unseen conditions or downstream OOD generalization. What is demonstrated is disentanglement under observed conditions. That is still useful, but narrower than the framing.

6. **Comparison fairness needs more transparency.** The appendix suggests that, at least in the scRNA-seq experiment, DisCoVR uses a larger `w` dimensionality than the baselines. If so, that is a meaningful capacity difference and should be normalized or explicitly justified. More generally, the hyperparameter-search protocol is described too briefly for a paper making strong superiority claims across several baselines. I would want clearer reporting of search ranges, validation criteria, and compute budget.

7. **There are a few presentation/technical inconsistencies that should be corrected.**
   - The statement “sufficiency of the shared latent representation” is attached to conditioning on `w`, not `z`, which is conceptually confusing.
   - Algorithm 1 appears to use gradient descent for both the classifier and the encoder/decoder block even though the objective is written as a max-min game; at minimum the sign convention needs clarification.
   - The justification for using the classifier on `\hat{x}` rather than directly on `z` is weaker than the text suggests. A lower-capacity classifier on a deterministic function of `z` gives a weaker signal; it does not automatically justify the stronger information-theoretic interpretation in the paper.

8. **Compute is nontrivially higher for the proposed method in some settings.** The appendix shows that DisCoVR can be the most expensive model in runtime/memory, especially on CelebA. This is not disqualifying, but the paper should discuss the tradeoff more explicitly.

---

> ### Author Rebuttal · Authors · 2026-03-30
>
> Dear reviewer,
>
> Thank you for your detailed review.
>
> **W1 + Q1:**
> The proposition is a characterization of the idealized game at the function-space level, and serves to a justify the objective.
> You are correct that these assumptions do not hold for the implemented parameterization. The implemented method should be viewed as a standard relaxation (e.g., VAEs and their extensions) to this idealized formulation. We will make this distinction explicit.
>
> **W2 + Q2:**
> Yes, we assume only one unified probabilistic model. Given the joint model, $p(x \mid z)$ is obtained by marginalizing out $w$ and $y$, that is $p(x \mid z)=\sum_y \int p(x \mid z,w)p(w \mid y)p(y)dw$.
> For this marginal conditional, $L_z$ is exactly the standard ELBO for approximating the posterior $p(z \mid x)$.
>
> **W3 + Q5:**
> The coupling between $w$ and $z$ is not defined at the level of individual samples. The prior for $w$ depends only on the class-level mean of the $z$ embeddings, not on the specific latent $z_i$ of an observation $x_i$.
> We noticed a typo (a missing subscript in the expectation) that might have contributed to the confusion: in line 132 the expression should be $\mu_y = E_{p(x\mid y)}E_{q(z\mid x)}[z]$.
>
> In practice, these expectations are replaced by empirical averages: for each label $y$, we average $z_i$ over all samples such that $y_i=y$.
> The number of conditions and class imbalance affect this only indirectly, through whether there are enough samples for each value of $y$. However, if a class is too underrepresented to support a reliable estimate of this average, then disentanglement with respect to that condition is likely to fail regardless of the chosen prior.
>
> **W4 + Q4:**
>
> To address this, we performed the experiments you suggested (see below). However, unlike a direct evaluation of the representation, post hoc prediction from a representation depends on the choice and design of the downstream classifier, and therefore is not a robust measure of how much $y$-relevant information the representation retains. A clear example is the prediction of $y$ from $w$ in the CelebA results below.
>
> For this reason, we reported quantities that more directly address this question.
> In the synthetic experiments, $I(y;w)$, $I(y;z)$ and  $MIC(w;z) = \frac{I(y;w)}{I(y;w)+I(y;z)}$ can be computed explicitly and are reported in App. E.1 and E.2.
>
> For other experiments, we inspect reconstructions from $z$ alone and from $w$ alone. For example, in colored-MNIST, reconstructions from $z$ are uniformly purple, indicating that they do not retain whether the original digit was red or blue. In CelebA, reconstructions from $z$ show "semi-glasses," so they do not preserve clear information about whether the original face wore glasses.
>
> Classification accuracy for $z \rightarrow y$:
>
> | Method | Swiss Roll | Colored MNIST | scRNA-Seq | CelebA |
> |---|---:|---:|---:|---:|
> | KNN | 0.518 | 0.546 | 0.594 | 0.571 |
> | Random Forest | 0.520 | 0.568 | 0.604 | 0.512 |
> | MLP | 0.506 | 0.629 | 0.597 | 0.611 |
>
> Classification accuracy for $w \rightarrow y$:
>
> | Method | Swiss Roll | Colored MNIST | scRNA-Seq | CelebA |
> |---|---:|---:|---:|---:|
> | KNN | 0.628 | 0.764 | 0.994 | 0.621 |
> | Random Forest | 0.696 | 0.786 | 0.993 | 0.556 |
> | MLP | 0.696 | 0.797 | 0.993 | 0.831 |
>
> **W5:**
> Domain generalization and OOD appear as motivation in the first paragraph of the introduction, and again in Appendix A when discussing broader connections to related work. The paper is explicit that its goal is to learn disentangled representations, for example in lines 39--40, 56--57, and 67--68.
>
> **W6 + Q3**:
> Indeed, only in the scRNA-seq experiment, our model used a larger $w$ dimension. As explained in the appendix, this was a simplifying modeling choice. We agree, however, that it introduces an unnecessary asymmetry in the comparison, and we   rerun the experiment with matched dimension. The updated results are qualitatively the same (we will replace the figures too):
>
> |w - Stimulation ($\uparrow$) | z - Cell Type ($\uparrow$) | z - Stimulation ($\downarrow$)
> |--|--|--|
> | 0.906 +- 0.003 | 0.716 +- 0.031 |0.002 +- 0.001|
>
> **W7:**
> $ I(\hat{x}, y)$ is a lower bound on $I(z, y)$ by the data processing inequality, so minimizing it discourages class information from being retained in $z$. Nevertheless, Appendix F includes an ablation study showing that applying the discriminator directly to $z$ leads to the same results as $\hat{x}$, but requires substantially more parameter tuning.
>
> **W8:**
> Indeed, particularly in Celeb-A, our method was more expensive. We are happy to make this more explicit.
>
> **Limitations:**
> Please note that the "limitations" section in the reviewers form is regarding **ethical limitations**. Our statement is the official ICML-recommended one for general machine learning papers with no particular societal concerns.
> Regardless, the mentioned ones are addressed in our responses above.

---

### Official Review · Reviewer_r5Pj · 2026-03-12

**Soundness:** 3
**Presentation:** 3
**Significance:** 2
**Originality:** 3
**Overall Recommendation:** 4
**Confidence:** 3

**Summary:**

This paper proposes a VAE framework utilizing dual latent codes to learn disentangled representations through a max-min strategy. The authors detail the theoretical design of the generative model and provide a derivation of the corresponding ELBO for the disentanglement objective. The proposed method is evaluated across multiple benchmark datasets and achieves consistent improvement in disentanglement in comparison to the baselines.

**Compliance With Llm Reviewing Policy:**

Affirmed.

**Final Justification:**

Overall, this is a good paper with sound methods and a solid evaluation. The authors' response addressed my concerns.

**Key Questions For Authors:**

(1) In Proposition 2.1, how is the expectation over q_(w|x, y) implemented? Specifically, how is it handled when w is learned from classes, if I understand it correctly?

(2) What is the Mutual Information Completeness (MIC) metric? Is a lower or higher value considered better?

**Limitations:**

yes

**Strengths And Weaknesses:**

Strengths

(1) The theoretical proof and derivation is detailed. Overall they look solid and I carefully checked the ELBO.

(2) Both qualitative and quantitative (with multiple metrics) evaluation demonstrate the proposed method archives a good disentanglement of the class-dependent and -independent factors.

Weaknesses

(1) The authors are encouraged to conduct a more comprehensive evaluation of the learned representations. For example, what specific representations are captured by the class-dependent versus the class-independent components? Visualizing latent traversals for both components separately would be helpful for readers to understand the semantic usefulness of the learned representations, especially for the class-independent part.

---

> ### Author Rebuttal · Authors · 2026-03-29
>
> Dear reviewer,
>
> Thank you for your review.
>
> **Regarding class-dependent versus the class-independent representations:**
>
> Complex high-dimensional data such as images typically requires high-dimensional latent representations. Such representations are inherently difficult to visualize directly. A meaningful proxy however, is the reconstructions they support.
>
> For this reason, for example, in the colored-MNIST experiment (Fig. 4, with additional analysis in Supplementary Figs. 2--4) and the CelebA experiment (Fig. 5, with additional analysis in Supplementary Fig. 5), we visualize reconstructions obtained from $z$ alone (label-agnostic) and from $w$ alone (label-aware). These visualizations are consistent with the intended roles of the two components:
>
> - In Fig. 4, $w$ captures color, while $z$ is encouraged to be color-invariant. Accordingly, reconstructions from $z$ appear in a mixed red-blue, or purple, tone, indicating that color-specific information has been removed.
>
> - In Fig. 5, the middle row shows reconstructions from $w$ alone. Since $w$ is conditioned on the presence or absence of glasses, these reconstructions match the top row with respect to that attribute. The bottom row shows reconstructions from $z$ alone. Because $z$ must represent the image while remaining invariant to glasses, so it produces "semi-glasses" for all faces.
>
> An exception to this difficulty is simulated data where we know that a low-dimensional representation is sufficient. Then, the latent space itself can be visualized directly. We do so in Fig. 3 for the Swiss roll simulation, which shows disentanglement of shape and color in the representation space. Supplementary Fig. 2 further presents posteriors reconstructed from these representations. Similar analysis for the intervened biological data is provided in Fig. 6 and Supplementary Fig. 6.
>
> Even more so, where possible, we directly measure mutual-information based metric, measuring how much $y$-relevant information each representation carries. These are reported in Tables 4-9. They show that indeed $z$ does not encode $y$-relevant information, while $w$ does.
>
> **Answers to questions:**
>
> 1. $q_{w \mid x,y}$ is the learned posterior distribution of the condition-dependent latent representation. As in other VAE based models, the expectation with respect to this posterior is implemented by Monte Carlo approximation, that is, by averaging over samples drawn from $q(w \mid x,y)$. In our model, this means sampling from a Gaussian.
> This is described in Section 3  in lines 190--194 in the left column, and the reparameterization used to obtain differentiable samples is given in lines 211--218 in the right column. We will emphasize this further.
>
> 2. The Mutual Information Completeness (MIC) metric is defined in Appendix E (line 1048). In our setting, it measures how much of the information about $y$ is captured by the condition-dependent representation $w$, relative to the total amount of $y$-information captured by both $w$ and $z$. MIC is high when $I(y;w)$ is large and $I(y;z)$ is small. Therefore, values closer to $1$ are better, as they indicate that most of the information about $y$ is concentrated in $w$, with little ``leakage" into the condition-independent representation $z$.

---

> > ### Author Rebuttal · Reviewer_r5Pj · 2026-04-02
> >
> > I thank the authors for their detailed response, and I believe my concerns have been addressed.

---

### Decision · Program_Chairs · 2026-04-30

**Decision:**

Accept (regular)

**Comment:**

The reviewers appreciated the novelty, method design and thorough experiments. Raised conserns were address by the rebuttals as aknowledged by the reviewers, in case they responded, or according to my judgment otherwise. I recommend accpetance for a good quality paper for the subcommunity of variational inference.